# LINK PREDICTION ON TEXT ATTRIBUTED GRAPHS: A NEW BENCHMARK AND EFFICIENT LM-NESTED GRAPH CONVOLUTION NETWORK DESIGN

## ABSTRACT

Textual and topological information is significant for link prediction (LP) in text-attributed graphs (TAGs). Recent link prediction methods have focused on improving the performance of capturing structural features by Graph Convolutional Networks (GCNs). The importance of enhancing text encodings, powered by the advanced Pre-trained Language Models (PLM) has been underestimated. In this work, we analyse and emphasise the importance of PLMs and propose a novel PLM-nested GCN design. We developed an extensive benchmark to compare current competitive link prediction methods and PLM-based methods in a unified experimental setting and systematically investigate the representation power of the text encoders in the link prediction task. Based on our investigation, we introduce LMGJOINT — a memory-efficient fine-tuning method. The key design features include: residual connection of textual proximity, a combination of structural and textual embeddings and a cache embedding training strategy. Our empirical analysis shows that these design elements improve MRR by up to 19.75% over previous state-of-the-art methods and achieve competitive performance across a wide range of models and datasets.

## 1 INTRODUCTION

Link Prediction (LP) aims to predict the likelihood of a connection between two nodes in a graph, encompassing various real-world applications, including protein-protein interaction prediction (Szklarczyk et al., 2018), recommendation systems (Huang et al., 2005) and knowledge graph completion (Hu et al., 2020b). While early LP relied on handcrafted graph heuristics (Adamic & Adar, 2003), more advanced approaches follow a two-stage framework: (1) an encoder maps graph information into node embeddings and (2) then a decoder assesses pairwise embedding similarity to predict connection likelihood.

Among encoder designs, Graph Convolutional Networks (GCNs) are the dominant paradigm, depending on both node and structural features. In previous benchmarks such as Cora (McCallum et al., 2000) and PubMed (Sen et al., 2008b), node features have often relied on shallow text embeddings such as Word2Vec (Mikolov et al., 2013). However, these embeddings struggle to capture context-aware information and complex semantic relationships, which are crucial for link prediction.

Despite their limitations, these shallow embeddings are widely used in standard benchmarks (Hu et al., 2021), which has led to several issues: (1) they are often practically irreproducible, making it difficult to replicate them on new datasets; (2) the reliance on a specific text embedding has resulted in architecture over-optimization in current algorithm development; and (3) the decoupling of the text embedding process and the GCN design hinders seamless end-to-end training, thereby reducing the overall effectiveness of the approach.

Text-attributed graphs (TAGs) help overcome these limitations by offering rich semantic content. They enable the characterization of individual node properties using powerful Pretrained Language Models (PLMs). Additionally, TAGs allow for the seamless integration of learnable text embeddings with GCN-based structural aggregation. However, existing works on TAGs primarily focus on node classification (Duan et al., 2024; He et al., 2023; Yang et al., 2021; Zhu et al., 2024b) or suffer from limited comparisons due to a lack of strong baselines (Wang et al., 2023; Yun et al., 2021;

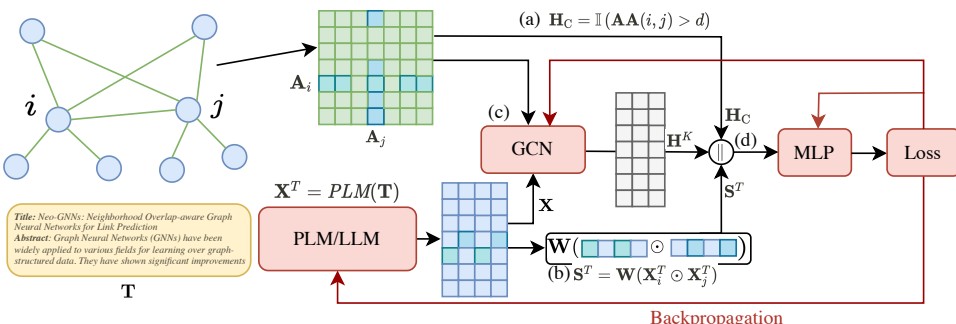

Figure 1: The overview of LMGJOINT. The framework consists of three main components: (a) structure embedding $\mathbf{H}_C$ (soft/hard common neighbors) from the adjacency matrix $\mathbf{A}$. (b) $\mathbf{S}^T$ semantic embeddings proximity based on sentence embedding derived from PLM. (c) Aggregated embeddings $\mathbf{H}^K$ which incorporate both $\mathbf{X}^T$ and $\mathbf{X}$. (d) The final step concatenates (a,b,c) through a MLP to generate link prediction.

Chamberlain et al., 2023; Zhang et al., 2020; Zhang & Chen, 2018). Motivated by this limitation, we make the following contributions:

1. **Data contribution** We collect and introduce ten graphs including Small PaperwithCode (Saier et al., 2023), Cora (McCallum et al., 2000), Arxiv_2023 (He et al., 2023), PubMed (Sen et al., 2008b), Medium PaperwithCode (Saier et al., 2023), Photo Shchur et al. (2018), History (Li et al., 2024), Ogbn-arxiv (Hu et al., 2021), Citationv8 (Wu et al., 2021) and Ogbn-papers100M (Hu et al., 2021). We provide rich statistics compactly describing their density, hierarchy, locality and generalized node homophily. These datasets and statistics serve as a foundation for exploring these new hypotheses driving the link prediction community moving forward.

2. **Extensive Empirical Benchmark** Using the proposed datasets, we have provided a thorough benchmark, offering a fair comparison of ten GCN-based link prediction approaches alongside seven traditional path-based methods. These selections broadly represent the current LP algorithm space, including state-of-the-art methods. Additionally, we expand the PLM-based baselines by adapting analogous architectures from node classification. This includes both cascade and nested architectures, as discussed in Section 5. Our benchmark is available at TAG4LP.

3. **LMGJOINT, a powerful nested framework** We introduce **L**anguage **M**odel **G**raph **J**oint Design (LMGJOINT), a parameter- and memory-efficient method. We identify three key design features, including (D1) residual connection of textual proximity, (D2) combination of structural and textual embeddings and (D3) cache embedding training strategy. We provide a theoretical justification for these design principles in Section 4. Our integration results in a fine-tuned architecture that preserves the GCN's strength in structural feature extraction while leveraging the PLM's ability to capture complex semantic relationships. Experimental comparisons with state-of-the-art approaches demonstrate that LMGJOINT achieves up to a 19.75% improvement in MRR. Moreover, experiments across seven proposed datasets, scaling up to $10^7$ nodes, validate the effectiveness and scalability of our approach, consistently outperforming competitive baselines. Furthermore, LMGJOINT is not limited to specific GCNs. It can be easily combined with any graph-based model and PLM-based text encoder without requiring any changes to the latter or affecting its computational complexity.

## 2 RELATED WORK

**LM-based approaches for TAGs.** Shallow embedding: In the context of TAGs, previous preprocessing methods often involved transforming text attributes into bag-of-words (Harris, 1954). Though widely adopted in the graph community, it has limited capacity to capture complex semantic relationships or fully utilize the richness of text attributes provided by modern pre-trained language models (PLMs). PLM-based method: To address these limitations, recent approaches leverage fine-tuning of

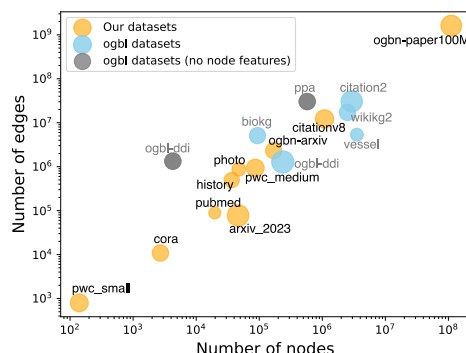

Figure 2: The figure compares TAG4LP (yellow) with previous ogbl datasets (blue) (Hu et al., 2021). Circle sizes indicate generalized edge homophily (Eq 44), with non-featured graphs (gray) set to 0.5. For more details, see Appendix G

pre-trained LMs to generate node embeddings tailored to the domain and context of TAGs. It can be classified into two main frameworks: (1) Cascade framework: Text embedding from PLM and graph aggregation are performed sequentially without interaction. Examples include SimTeG (Duan et al., 2024), GAINT (Chien et al., 2021) and TAPE (He et al., 2023). (2) Nested framework: PLMs and GCNs are optimized jointly, enabling iterative or integrated learning. For instance, Graphformer integrates text encoding and graph aggregation in an iterative workflow (Yang et al., 2021), while Engine incorporates caching and a dynamic early-exit mechanism to enhance performance and reduce training costs in Llama 3 (Zhu et al., 2024b). A detailed comparison of our benchmark with related work is provided in Table 1. (3) Instruction Learning: GraphGPT (Tang et al., 2023) integrates LLMs with graph structural knowledge through instruction tuning. Furthermore, LinkGPT (He et al., 2024) proposes a two-stage fine-tune method, achieving state-of-the-art performance in zero-shot and few-shot settings.

**Related Benchmarks** Our proposed method bears methodological resemblance to (Zhang et al., 2024a). It benchmarks a co-training method on 22 graphs in both link prediction and node classification tasks. However, the proposed approach fails to bring performance gain for link prediction. Besides, Mao et al. (2023) critically examines the fundamental incompatibility between node features and structural similarity, which grounds the analysis from a data science perspective. Li et al. (2023) proposes a benchmark of all existing GCN4LP methods under consistent data splits and training settings. Their findings reveal that advancements in GCN4LP are primarily due to improved capture of pairwise structural features. Similarly, Wu et al. (2021) introduces a new TAG benchmark, mainly focusing on node classification. It proposes a co-training paradigm by simply concatenating GNNs and LLM/PLMs several cascade-architectures without a task-specialised design. Recent work starts considering including edge textual features into TAG and conducting various experiments on cascade GCN-LLM models (Li et al., 2024). In summary, while current methods for node classification have provided foundational insights into pretraining tasks and algorithm design, there is no counterpart and established SOTA for link classification tasks.

**Related Datasets** The widely used datasets for link prediction (LP) were introduced by OGB (Hu et al., 2021), but their rich textual attributes have been largely underexplored. Recently, TAPE (He et al., 2023) and TAG_Benchmark (Yan et al., 2023) introduced several text-attributed graphs (TAGs), such as Cora (McCallum et al., 2000), PubMed (Sen et al., 2008b), and Arxiv_2023 (He et al., 2023). The Engine further expanded these datasets into seven graphs. Similarly, Chen et al. (2024b) conducted preliminary studies on three datasets for LP. It observes that without task-specific design simply combining LLM and GCN in a nested architecture fails to achieve performance improvements. Recently, edge-level textual features have garnered attention, with Li et al. (2024) introducing a benchmark of 9 graphs. However, this benchmark is limited to a cascade GCN-LLM design. Building on insights from these works, our benchmark focuses initially on homophilic networks (Hu et al., 2021) and later generalizes to other domains and non-attributed graphs. Key distinctions from existing benchmarks are highlighted in Figure 2. In summary, our benchmark is the most comprehensive, featuring the widest variety of algorithms and the largest number of datasets evaluated.

## 3 DATASET CONTRIBUTION

**Data factors and Current Limitation**: The further development of the Link Prediction algorithm is hindered by the efficient hypothesis. The limitations of applying GNNs for node classification on heterophily graphs are well understood. In comparison, prior works on GNN4LP are mostly based on hand-crafted structure features (Zhang et al., 2020; Zhang & Chen, 2018; Wang et al., 2023; Yun et al., 2021). Despite the practical improvement, our understanding of the dominant data factor within GNN4LP remains incomplete. We identify three critical data factors: 1) *Feature homophily* refers to the impact of similar features, a recent study indicates discrepancies between feature proximity and structure Zhu et al. (2024a) leads to performance decay for link

Table 1: Comparison with existing methods. ✓: public benchmark, ✓: benchmarked model, NC: node classification, LP: link prediction, Num: number of the evaluated dataset

| Task | Works | Cascade | | Nest | | Num |
|------|-------|---------|-----|------|-----|-----|
| | | MLP | GCN | MLP | GCN | |
| NC | Graphformer | | | | ✓ | 3 |
| NC/LP | GAINT | ✓ | ✓ | | | 2 |
| NC/LP | SimTeG | ✓ | ✓ | | | 3 |
| NC | TAPE | | ✓ | | | 5 |
| NC | LEADING | | ✓ | | | 3 |
| NC | ENGINE | | | | ✓ | 7 |
| NC | TEG-DB | ✓ | ✓ | | | 9 |
| LP | Ours | ✓ | ✓ | ✓ | ✓ | 9 |

prediction. We quantify this data factor by generalized edge homophily defined in Appendix G 2) *Structure hierarchy* describes the hierarchical structure that widely exists in the citation network. When embedding such a graph in Euclidean space, GCN-based embedding incurs a large distortion compared to in hyperbolic space Liu et al.; Chami et al. (2019). We quantify such hierarchy using $\alpha$ in degree distribution; 3) *Pairwise local structure*: This hypothesis originates from the intrinsic permutation invariance of GCN. It results in the limited expressivity to distinguish automorphic nodes Chamberlain et al. (2023). To analyze and study their impact on link prediction from a data-centric perspective, we suggest clustering and transitivity to measure such local distance features. To sum up we propose 12 graph statistics, covering three categories, as shown in Table 5. These statistics compactly and thoroughly quantify the above-mentioned three data factors. Our proposed dataset and statistics provide valuable resources to advance research in the TAG and GRL communities. Dataset statistics can be found in Appendix G.3.

**Proposed dataset** To address the above limitations, we introduce a novel TAG dataset comprising eight graphs from prior literature and two generated graphs. This collection offers several distinct advantages: (1) *Expanded Scale*: Our collection builds on widely adopted benchmarks in the graph research community, such as Cora, PubMed and Arxiv_2023 to ensure consistency and comparability with existing studies. Additionally, we introduce two datasets derived from the PaperswithCode API (Saier et al., 2023). It provides a continuous spectrum of node sizes ranging from $10^2$ to $10^9$. To further enhance scalability and enable the study of large-scale settings, we include Citationv8 (Yan et al., 2023) and ogbn-paper100M (Hu et al., 2021). (2) *Enriched Textual Information*: Except traditional node features derived from shallow embedding (Mikolov et al., 2013; Harris, 1954), our dataset also retains the original textual content associated with nodes. This enriched textual information enables more algorithmic flexibility to provide more advanced text encoding using PLMs.(3) *Extensive and Comprehensive Statistics*: Previous datasets have typically reported only the number of nodes and edges, offering limited insights into the underlying structural complexities. We offer a richer set of statistics in Appendix G relates to current hypothesis, including density, hierarchy, locality, and feature homophily (Zhu et al., 2024a). We illustrate the difference between proposed dataset with OGB (Hu et al., 2021) in terms of scale and feature homophily in Figure 2. More details about data statistics can be found in Appendix G.3.

## 4 LMGJOINT: A NEW EFFICIENT MODEL

We begin by describing three basic components (C1, C2, C3) that guided our design and then highlight three efficient designs (D1, D2, D3) that can help improve the performance and reduce memory requirements.

**C1: Soft Common Neighbor**: On the other extreme, we utilize Common Neighbor, a first-order structure feature that solely utilizes graph topology.

$$\mathbf{H}^{\mathrm{C}}_{ij} = \mathbb{I}\left(\mathbf{A}\mathbf{A}_{(i,j)} > d\right) \tag{1}$$

where $\mathbf{A} \in \{0, 1\}^{n \times n}$ is the binary adjacency matrix , $\mathbb{I}$ is indication function, $d$ is a hard threshold to remain stronger connections. Common neighbors can also be leveraged to directly detect the likelihood of a connection between nodes, i.e. Hard Common Neighbors (Newman, 2001).

**C2: Semantic Feature Proximity**: In a homophilic setting, connected nodes exhibit high textual proximity. Thus, a straightforward approach is to disregard graph structure and train a multilayer perceptron (MLP) solely on the text encodings. Let $\mathbf{T}, \mathbf{X}^T$ represent the raw text and embedded text features from PLM. The semantic proximity between node pair $(i, j)$ is defined as:

$$\mathbf{X}^T = \text{PLM}(\mathbf{T}) \tag{2}$$

$$\mathbf{S}_{ij}^T = \mathbf{W}(\mathbf{X}_i^T \odot \mathbf{X}_j^T) \tag{3}$$

Here $\mathbf{W}$ is a learned weight matrix. The operator $\odot$ denotes the Hadamard product. PLM is the pre-trained embedding model that maps raw text to a numeric vector. We benchmark three different sentence embedding methods including e5-large-v2 (Wang et al., 2022), Sentence-Transformers MiniLM-L6-v2 (Reimers & Gurevych, 2019a) and MPNet(Song et al., 2020).

**C3: Aggregated Semantic Feature with Self-loop**: The aggregated features are propagated with a self-loop to capture the information of k-step neighbors. This is useful to capture the information of similar neighbors when the structure exhibits homophily e.g. in a citation graph (Lee et al., 2024).

$$\mathbf{H}^k = f\left(\tilde{\mathbf{A}}_{\text{sym}} \mathbf{H}^{k-1} \mathbf{W}\right) \tag{4}$$

$\mathbf{H}^0 = \mathbf{X}$ and successively optimized by $\mathbf{X}^T$ from PLM by cache embedding strategy introduced in Section 4.1. We symmetrically normalize the adjacency matrix $\tilde{\mathbf{A}}_{\text{sym}} = (\mathbf{D} + \mathbf{I})^{-\frac{1}{2}}(\mathbf{A} + \mathbf{I})(\mathbf{D} + \mathbf{I})^{-\frac{1}{2}}$, $\mathbf{I}, \mathbf{D}$ are the identity and diagonal degree matrix (Kipf & Welling, 2016). We can collapse the repeated multiplication with the normalized adjacency matrix $\tilde{\mathbf{A}}_{\text{sym}}$ into a single matrix to the K-th power, $\tilde{\mathbf{A}}_{\text{sym}}^K$. Then we have the aggregated feature as $\mathbf{H}^K = f\left(\tilde{\mathbf{A}}_{\text{sym}}^K \mathbf{X} \mathbf{W}\right)$.

## 4.1 EFFICIENT NESTED ARCHITECTURE

**Ours: LMGJOINT** We combine embeddings from these three simple components through simple linear transformations and component-wise non-linearities Lee et al. (2024); Wang et al. (2023); Chamberlain et al. (2023).

$$\mathbf{Y} = \text{MLP}\left([\mathbf{H}^K; \beta \mathbf{H}_C; \mathbf{S}^T]\right) \tag{5}$$

$\mathbf{Y}$ is our model's output predictions. $\beta$ is the weight for the structure feature. In Fig. 1 we visualize LMGJOINT. We also give its pseudocode in Algorithm 1. We then

**D1: Jumping Connection of Textual Similarity** GNNs primarily depend on node-level aggregation via summing the weighted neighboring features iteratively. Such a local smoothing mechanism helps generate more representative embedding when the homophily assumption holds Luan et al. (2023). However, it also smooths the rich semantic embedding from textual nuance during the smoothing process. We address such issues by combining pairwise semantic proximity without training at the last layer, i.e., the semantic similarity representations "jump" to the last layer Xu et al. (2018b). A jump connection that bypasses the GCN, directly transmitting feature proximity (textual similarity) to the final embeddings, as illustrated in Figure 1 (b). Additionally, we provide a theoretical justification for this design from an information-theoretic perspective in Appendix A.1.

**D2: Combination of Structural and Semantic Embeddings** The interaction between feature proximity and structural proximity dictates the formation of links. Previous theoretical work has demonstrated that local structural proximity and feature proximity are often incompatible, i.e. node pairs with a large number of common neighbors tend to exhibit low feature proximity (Mao et al., 2023). Based on this insight, we simplify the current baseline model in Eq 5 to $\mathbf{Y} = \text{MLP}\left([\mathbf{H}^K; \beta \mathbf{H}_C + \mathbf{S}^T]\right)$ to reduce the hidden dimensions, thereby reducing time complexity during training. We proved in Appendix A, that for any node pair $(i, j)$, the approximation error of this design decreases as the number of nodes increases.

**D3: Cache Embedding Strategy** Figure 1 illustrates how gradients from the learning objective of LMGJOINT are back-propagated through both the GCN and the final encoder layers of the PLM. This integration allows the PLM to capture structural features while enabling the GCN to enhance feature

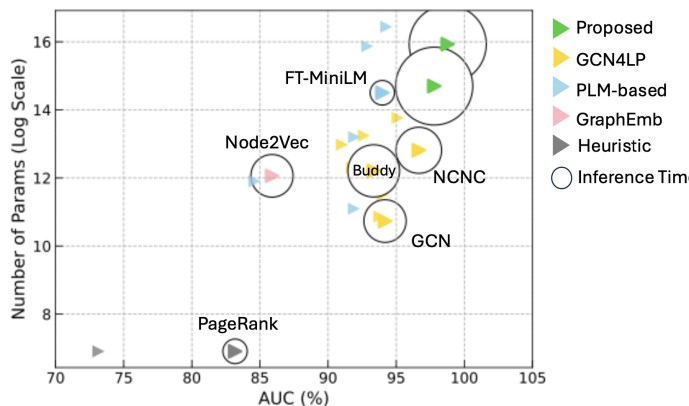

Figure 3: Illustration of the performance-complexity trade-off between LMGJOINT (green) and benchmarked methods on Cora. The x-axis represents AUC, the y-axis shows the number of parameters (log-scale) and the marker radius indicates inference time. Points closer to the bottom-right corner reflect higher cost-effectiveness.

aggregation. To mitigate the high memory cost of PLM training, we employ a cache embedding strategy. Node feature $H^0$ is initialized by default node features $X$, we save these pre-computed embeddings in a cache. In the current mini-batch, we re-encode only the tokens associated with the target and source links $(\mathbf{X}_i^T, \mathbf{X}_j^T)$ by PLM, and then concatenate them with pre-computed node features as the input for GCN $\mathbf{X} = [\mathbf{X}_{\mathcal{V}\setminus\{i,j\}}; \mathbf{X}_i^T; \mathbf{X}_j^T]$. This approach significantly reduces the per-mini-batch computational cost from $O(Nd)$ (where N is the number of nodes and d is the embedding dimension) to $O(d)$. In summary, our method is both parameter- and memory-efficient, enabling training on a single A100 GPU with 40GB VRAM across all proposed datasets. Detailed complexity analysis is provided in Appendix C.1.

While these principles have been employed independently in previous works (Zhu et al., 2020; Wang et al., 2023), we are the first to advocate for their combined necessity. We substantiate our claims with both theoretical justifications and a comprehensive empirical analysis across diverse datasets.

## 5 BENCHMARKING

In this section, we provide an overview of the various benchmarked methods. Specifically, we evaluate structure-only approaches (heuristic, structure-based) and text-only methods (PLM-Inf-MLP, FT-PLM-MLP) to assess current advancements in each setting. We categorize all existing approaches into two broad classes: graph-based and PLM-based. The former emphasizes improving pairwise structural proximity, while the latter focuses on feature proximity.

### 5.1 GRAPH-BASED METHODS

This section evaluates graph-based methods which can be categorized into four groups: 1) **Graph Heuristic**: Local heuristic leverages modified shared neighborhoods, including Common Neighbor (CN), Adamic-Adar (AA), Resource Allocation (RA). Other global heuristics such as Katz, Shortest Path and symmetric PageRank(Adamic & Adar, 2003; Page et al., 1999) consider all paths between connected nodes. 2) **Embedding-based Methods**: It focus on proximity-preserving embedding methods to model neighborhood contexts via random walks, including DeepWalk, Node2Vec and LINE (Perozzi et al., 2014; Tang et al., 2015; Grover & Leskovec, 2016). 3) **Aggregation-based Methods**: GCNs aggregate information recursively from first-hop neighbors (Kipf & Welling, 2016; Velickovic et al., 2017). SAGE embeds self and neighboring nodes separately to handle heterophily (Hamilton et al., 2017; Zhu et al., 2020). GIN achieves the same expressivity as Weisfeiler-Lehman test (Xu et al., 2018a) by employing injective transformation and we perform a dot product with an MLP layer to handle the final embeddings. 4) **GCN4LP**: SEAL, BUDDY and ELPH introduces different labeling techniques and structure embeddings to address the automorphic nodes problem (Zhang et al., 2020; Zhang & Chen, 2018; Chamberlain et al., 2023). The latest GCNs augment

aggregated features by leveraging local structures, such as CNs to enhance performance. Notable examples include NCN/NCNC (Wang et al., 2023) and NeoGNN (Yun et al., 2021). Among graph-based methods, categories (1), (2) and (3) are structure-only methods since they do not utilize node features. To sum up, we include CN, AA, RA, Katz, Shortest Path, PageRank, DeepWalk and Node2Vec, GCN, GIN, SAGE, GAT, SEAL, NeoGNN, ELPH, BUDDY, NCNC, NCN, HLGNN.

## 5.2 PLM-BASED METHODS

We introduce and transfer four extended benchmarked frameworks from node classification task, each with progressively increasing resource requirements. This section provides a detailed overview of the PLMs/LLMs used and their corresponding embedding configurations. We investigate the following configurations: (1) **PLM/LLM as a Fixed Inference Model (PLM/LLM-Inf)**: The PLM/LLM operates in a frozen state, generating static embeddings that are then fed into a Multi-Layer Perceptron (MLP) for binary classification. Specifically, the final hidden states of the PLMs are utilized as text embeddings. (2) **Fine-Tuned PLMs (*FT-PLM*)**: Extending the **PLM/LLM-Inf** setup, fine-tuning is introduced by training the last few encoder layers alongside the MLP (Chen et al., 2024a). (3) *PLM/LLM-Inf-GCN*: This configuration first generates text embeddings from the PLMs as node features, after which a GCN is trained on the updated node features, but without further training on PLM/LLM. (4) **FT-PLM-GCN**: Building upon (3), this approach optimizes all parameters, including those in the last encoder layers of the PLM, the GCN and the MLP (Yang et al., 2021). Configurations (1) and (2) are classified as text-only methods. To sum up, we benchmark cascade (3) and nested architecture (4) adopted from prior work in node classification.

**Selection of PLM/LLMs** In this paper, we define PLMs as those models practical for inference and fine-tuning within typical academic budgets, such as BERT (Devlin et al., 2019) and LLMs refer to models requiring substantial computational resources to fine-tune such as thousands of GPUs or TPUs, exemplified by Llama-3-8B (Dubey et al., 2024). We utilize both encoder-only and decoder-as-embedder (BehnamGhader et al., 2024) models for text embedding, including (1) BERT, a lightweight deep text embedding model pretrained in a self-supervised manner (Devlin et al., 2019); (2) e5-large-v2 (Wang et al., 2022), Sentence-Transformers MiniLM-L6-v2 (Reimers & Gurevych, 2019a) and MPNet(Song et al., 2020), pretrained using a contrastive learning approach. Additionally, we incorporate Meta-LLaMA-3-8B (Dubey et al., 2024),a decoder-only LLM, which we include as a case study for text embedding at scale. To ensure consistency and comparability with existing studies, we include shallow embedding methods such as bag-of-words (Harris, 1954) and Word2Vec (Mikolov et al., 2013). We leverage the [EOS] token in LLaMA3 and the [CLS] token in sentence embedding models as node features and fine tune them with full-parameter tuning strategy. Further details on fine-tuning and embedding strategy are provided in Appendix B.3. To sum up, we include BERT, e5-large-v2, MiniLM-L6-v2, MPNet and Meta-LlaMA-3-8B, bag-of-words and Word2Vec as text encoder with 4 experiment settings.

## 5.3 EXPERIMENT SETTINGS

**Metrics choice** To ensure consistency in previous works Hu et al. (2021), we benchmark all approaches on the Cora, PubMed and Arxiv_2023. Furthermore, our evaluation is extended to all nine datasets with metrics including Hits@50, Hits@100, MRR (Mean Reciprocal Rank) and AUC. Hits@K quantifies the ratio of positive edges ranked within the top K positions, while MRR evaluates the model's ability to rank positive above negative ones. AUC assesses the model's ability to score positives higher than negatives, offering numerical stability and scale invariance (see Appendix D). To avoid distribution shift caused by the feature-based split method (Wang et al., 2023), we apply a uniform random split of 80%, 15% and 5% across all datasets. For all experiments, the results are reported on randomly sampled test edges for datasets larger than pwc_medium in Table 3. All metrics are reported as the mean and standard deviation, averaged over five random seeds. We exclude the target link (link to be predicted) in each mini-batch.

**Hyperparameter Ranges** We utilize hierarchical grid search for hyperparameter optimization across all GCN models, tuning parameters such as learning rate, weight decay, the number of convolution layers, MLP layers, the number of heads in GAT and hidden neurons. For details on specific hyperparameters in GCN4LPs. Due to the training burden for PLM/LLMs, we use the same parameters for GCN and GCN4LPs in LLM-GCN related methods (i.e., PLM/LLM-Inf-GCN, FT-PLM-GCN and Ours) and only optimize the learning rate and weight decay. We utilize the same

Table 2: Benchmark results showing mean $\pm$ stdev for Hits@50, Hits@100, and MRR metrics on Cora, PubMed, and Arxiv_2023. The top 1-3 ranked models are highlighted in emerald, while ranks 4-6 are highlighted in green. Darker colors represent higher ranks. All results are provided under our benchmark with consistent training and evaluation settings. Heuristic, Structure, GCN, and GCN4LP are existing approaches, followed by extended baselines, with the final category representing proposed methods. $\boxed{\text{model}}$ highlights the best model among the existing approaches. We integrated both soft and hard CN into our LMGJOINT. To distinguish, LMGJOINT and LMGJOINT-C refers to soft and hard common neighbor introduced in Section 4.

| Category | Models | Cora | | | PubMed | | | Arxiv 2023 | | |
|---|---|---|---|---|---|---|---|---|---|---|
| | | Hits@50 | Hits@100 | MRR | Hits@50 | Hits@100 | MRR | Hits@50 | Hits@100 | MRR |
| Heuristic | CN | 50.36±0.03 | 50.36±0.03 | 32.88±0.09 | 33.32±0.02 | 33.32±0.02 | 21.13±0.02 | 27.20±0.01 | 27.20±0.01 | 14.66±0.06 |
| | AA | 50.36±0.03 | 50.36±0.03 | 47.33±0.09 | 33.32±0.02 | 33.32±0.02 | 24.61±0.11 | 27.20±0.01 | 27.20±0.01 | 19.87±0.30 |
| | RA | 50.36±0.03 | 50.36±0.03 | 47.17±0.11 | 33.32±0.02 | 33.32±0.02 | 23.94±0.16 | 27.20±0.01 | 27.20±0.01 | 19.16±0.27 |
| Structure | PPR/sym | 84.74±0.00 | 88.93±0.00 | 58.86±0.98 | 69.81±0.02 | 72.95±0.01 | 28.04±0.91 | 65.68±0.02 | 67.86±0.02 | 26.57±0.82 |
| | Katz | 69.25±0.02 | 69.25±0.02 | 38.17±0.12 | 66.02±0.02 | 66.02±0.02 | 30.94±0.08 | 55.39±0.01 | 55.39±0.01 | 21.76±0.21 |
| | DeepWalk | 84.00±0.07 | 89.31±0.03 | 44.39±0.96 | 64.35±0.01 | 71.78±0.01 | 19.66±1.00 | 21.37±0.11 | 33.80±0.10 | 4.47±0.03 |
| | Node2Vec | 83.08±0.05 | 88.38±0.03 | 39.94±1.10 | 63.95±0.02 | 70.95±0.01 | 20.68±0.24 | 30.19±0.18 | 40.81±0.20 | 5.60±0.03 |
| GCNs | GCN | 91.46±2.36 | 96.20±1.71 | 45.84±8.40 | 83.11±2.19 | 89.59±1.73 | 24.55±4.02 | 45.07±0.87 | 52.46±1.44 | 17.62±3.34 |
| | GAT | 89.80±2.00 | 95.34±1.61 | 49.82±10.04 | 74.23±2.54 | 84.03±1.59 | 18.13±5.81 | 43.09±1.22 | 53.49±1.06 | 13.58±4.33 |
| | SAGE | 86.40±3.73 | 95.10±1.57 | 46.03±6.70 | 86.55±0.55 | 93.36±0.70 | 35.63±5.75 | 45.42±3.12 | 56.69±3.13 | 11.52±1.67 |
| | GIN | 91.54±2.91 | 96.05±2.13 | 51.90±6.65 | 86.92±1.68 | 92.02±0.91 | 24.63±2.24 | 45.35±2.58 | 53.22±2.05 | 14.79±4.53 |
| GCN4LP | SEAL | 87.38±3.06 | 92.03±2.96 | 37.81±9.93 | 84.62±3.53 | 89.52±1.27 | 49.02±13.91 | 56.98±1.89 | 67.34±3.74 | 22.47±3.69 |
| | NeoGNN | 81.03±3.11 | 90.04±2.02 | 41.48±5.11 | 73.17±5.29 | 81.25±8.14 | 31.44±3.85 | 64.54±11.14 | 69.34±8.56 | 28.07±15.62 |
| | ELPH | 87.30±4.94 | 94.91±2.17 | 39.86±10.20 | 59.19±5.58 | 74.62±1.64 | 24.61±3.17 | 57.66±1.55 | 66.95±3.62 | 29.22±5.95 |
| | BUDDY | 87.82±3.14 | 95.42±2.26 | 30.78±5.55 | 76.14±3.46 | 89.25±2.27 | 19.46±2.42 | 52.25±2.01 | 60.49±0.94 | 18.75±3.71 |
| | HL-GNN | 90.59±3.41 | 94.62±1.87 | 50.35±10.07 | 85.14±1.83 | 91.87±1.36 | 31.49±7.84 | 76.51±1.68 | 84.23±0.82 | 24.21±7.69 |
| | NCN | 96.16±1.62 | 98.74±0.96 | 45.76±16.39 | 86.44±2.03 | 93.21±1.10 | 25.92±4.33 | 82.34±2.45 | 88.83±1.43 | 37.92±13.21 |
| | NCNC | 95.42±2.41 | 98.67±0.76 | 48.68±18.60 | 86.49±0.99 | 93.74±0.25 | 20.31±6.51 | 81.86±1.64 | 89.13±2.08 | 35.67±12.30 |
| PLM-Inf-MLP | BERT | 35.79±2.50 | 56.90±3.26 | 3.42±0.47 | 36.12±0.37 | 48.73±1.43 | 6.56±0.70 | 37.66±1.57 | 48.74±1.15 | 10.04±0.85 |
| | MiniLM | 83.39±0.00 | 92.99±0.00 | 34.29±4.10 | 66.35±0.29 | 81.90±0.03 | 21.54±0.11 | 68.15±0.09 | 77.62±0.03 | 16.91±0.18 |
| | e5-large-v2 | 64.35±1.56 | 83.10±0.80 | 24.40±2.48 | 71.32±0.86 | 82.59±0.26 | 21.79±1.58 | 75.03±0.28 | 84.09±0.24 | 21.69±0.03 |
| | Llama-3-8B | 89.15±0.72 | 95.64±0.41 | 31.19±3.49 | 79.87±1.19 | 89.01±0.53 | 22.87±4.47 | 83.18±1.19 | 89.91±0.19 | 22.85±1.12 |
| FT-PLM-MLP | BERT | 89.17±2.86 | 96.99±1.36 | 30.90±4.33 | 73.70±4.01 | 84.45±2.92 | 17.11±3.90 | 77.75±3.46 | 87.56±2.05 | 29.54±3.98 |
| | e5-large | 92.09±1.70 | 96.92±1.35 | 38.63±9.39 | 76.26±2.55 | 87.23±1.60 | 19.75±5.81 | 80.48±2.52 | 89.35±1.33 | 31.73±6.62 |
| | MiniLM | 92.49±2.13 | 96.68±1.69 | 35.55±5.82 | 75.87±3.72 | 86.80±1.98 | 20.79±6.32 | 80.20±2.62 | 88.38±1.06 | 29.86±5.82 |
| | mpnet | 93.44±1.64 | 97.78±0.66 | 22.55±10.71 | 63.27±31.76 | 90.69±2.49 | 9.38±3.12 | 82.72±1.28 | 91.44±0.75 | 8.42±6.49 |
| PLM-Inf-GCN | MiniLM-NCNC | 96.13±1.20 | 98.81±0.49 | 38.96±13.20 | 90.32±1.52 | 96.11±0.60 | 22.56±3.30 | 65.65±1.80 | 70.61±2.24 | 29.10±3.83 |
| | e5-large-NCNC | 96.13±1.13 | 98.81±0.74 | 39.23±12.99 | 90.86±1.95 | 96.69±0.56 | 27.02±5.96 | 83.24±1.20 | 90.46±1.14 | 25.14±9.39 |
| | Llama-NCNC | 95.57±1.02 | 98.73±0.65 | 27.45±7.86 | 84.65±1.95 | 92.39±1.46 | 20.51±9.80 | 84.68±1.72 | 91.90±1.33 | 27.16±11.48 |
| | BERT-NCNC | 77.47±1.77 | 84.11±2.70 | 25.39±12.42 | 72.80±1.78 | 82.48±1.43 | 23.49±3.07 | 58.83±3.91 | 68.50±3.49 | 22.80±2.55 |
| FT-PLM-GCN | MiniLM-GAT | 54.23±4.08 | 76.99±6.58 | 13.74±5.21 | 29.44±2.84 | 43.75±5.46 | 4.26±1.75 | 13.76±2.22 | 25.18±4.17 | 2.62±0.55 |
| | MiniLM-SAGE | 63.04±9.23 | 82.01±4.19 | 15.09±1.35 | 45.31±3.83 | 63.18±1.34 | 6.70±3.61 | 49.11±4.22 | 66.06±2.60 | 11.48±3.41 |
| | mpnet-GIN | 89.01±5.54 | 97.55±1.86 | 29.06±7.96 | 46.82±4.22 | 63.42±2.77 | 11.86±3.68 | 55.11±3.70 | 64.49±3.26 | 18.88±5.89 |
| | mpnet-GAT | 74.90±9.22 | 86.96±6.71 | 23.16±11.10 | 35.82±3.81 | 52.43±4.18 | 5.36±1.69 | 19.50±1.91 | 29.43±3.60 | 4.49±0.91 |
| | mpnet-SAGE | 82.01±4.19 | 93.88±0.28 | 25.34±8.06 | 57.58±3.78 | 71.62±2.78 | 11.91±3.58 | 52.97±5.05 | 66.28±4.50 | 14.51±3.37 |
| Ours | MiniLM-LMGJOINT-C | 99.92±0.18 | 99.92±0.18 | 41.52±19.50 | 99.91±0.09 | 99.94±0.08 | 44.99±10.82 | 90.61±2.25 | 98.16±1.73 | 35.47±10.91 |
| | MiniLM-LMGJOINT | 98.34±0.59 | 99.84±0.35 | 60.84±7.75 | 78.56±6.32 | 89.22±4.96 | 22.72±1.51 | 84.57±1.93 | 91.44±1.34 | 40.27±11.91 |
| | mpnet-LMGJOINT-C | 93.28±14.16 | 95.81±9.15 | 28.92±7.14 | 99.27±1.19 | 99.95±0.08 | 23.99±11.63 | 73.09±16.32 | 77.99±17.20 | 14.68±6.17 |
| | mpnet-LMGJOINT | 100.00±0.00 | 100.00±0.00 | 68.43±14.23 | 91.67±4.96 | 97.13±1.74 | 31.66±5.33 | 89.17±5.45 | 94.85±3.15 | 45.70±3.88 |
| | e5-large-LMGJOINT-C | 99.92±0.18 | 99.92±0.18 | 41.46±25.49 | 99.11±1.54 | 100.00±0.00 | 21.66±9.66 | 83.97±4.23 | 98.01±0.72 | 12.66±2.66 |
| | e5-large-LMGJOINT | 96.29±2.08 | 98.89±1.02 | 65.26±11.52 | 77.34±2.19 | 88.41±1.14 | 23.80±3.29 | 80.01±2.53 | 87.71±1.48 | 42.02±5.56 |

parameters of GCN and LLM in our proposed method. Further details about parameter tuning and experiment setting are provided in Appendix B.

# 6 BENCHMARK ANALYSIS

We analyze the benchmark results by addressing the following questions: (1) Is utilizing PLM alone more effective than a structure-based method? (2) Does the previous GCN4LP SOTA persist under the new configurations? (3) Should PLMs and GCN-based methods be trained separately?

**Text-only vs. Structure-only.** To address (1), we compare the performance between structure-only methods (Heuristic, Structure) and text-only approaches (PLM/LLM-Inf-MLP, FT-PLM-MLP) in Table 2. Although structure-only methods achieve better performance on Cora and PubMed when assessing MRR, text-only methods substantially show better performance in other metrics including Hits@50 and Hits@100. Furthermore, while assessing AUC as shown in Figure 3, the performance of FT-MiniLM, a text-only model, approaches the similar performance of a robust GCN4LP method NCNC. This suggests that PLM-based models can achieve strong performance even in the absence of topological information assessed Hits@K and AUC. However, it fails to outperform the structure-based method in MRR across benchmarked datasets under a unified experiment setting.

Table 3: Results on extensive datasets: Comparison with the strongest baseline in each category using AUC. Mean accuracy ± standard deviation is reported across different data splits. The best model for each benchmark is highlighted in emerald.

| | SMALL | | | MEDIUM | | | | LARGE | |
|---|---|---|---|---|---|---|---|---|---|
| | $\text{Pwc}_{small}$ | Cora | $\text{Arxiv}_{2023}$ | PubMed | $\text{Pwc}_{medium}$ | History | Photo | Ogbn-arxiv | Citationv8 |
| **Embedding-MLP: Non-contextualized Shallow Embeddings** | | | | | | | | | |
| TF-IDF | 63.50 ± 8.59 | 68.27 ± 2.52 | 76.65 ± 1.95 | 67.06 ± 2.34 | 70.94 ± 0.94 | 60.94 ± 0.48 | 62.80 ± 0.62 | 62.63 ± 2.88 | 57.18 ± 1.19 |
| Word2Vec | 51.00 ± 2.24 | 60.15 ± 1.77 | 85.22 ± 0.92 | 83.88 ± 0.91 | 81.79 ± 0.55 | 65.54 ± 0.21 | 64.71 ± 0.12 | 85.57 ± 0.28 | 80.50 ± 0.35 |
| **PLM-Inf-MLP: Local Sentence Embedding Models** | | | | | | | | | |
| MiniLM-L6-v2 | 51.90 ± 11.54 | 91.22 ± 0.04 | 95.22 ± 0.00 | 96.20 ± 0.00 | 98.39 ± 0.04 | 94.64 ± 0.01 | 84.14 ± 0.03 | 98.22 ± 0.01 | 97.86 ± 0.01 |
| e5-large-v2 | 80.60 ± 2.57 | 83.87 ± 0.23 | 95.72 ± 0.01 | 96.73 ± 0.03 | 97.83 ± 0.01 | 95.97 ± 0.00 | 85.04 ± 0.44 | 97.92 ± 0.01 | 98.05 ± 0.01 |
| BERT | 69.85 ± 2.40 | 65.09 ± 1.41 | 86.37 ± 0.27 | 88.96 ± 0.31 | 83.85 ± 0.33 | 90.26 ± 0.36 | 73.12 ± 0.76 | 86.89 ± 0.26 | 86.22 ± 0.48 |
| **LLM-Inf-MLP** | | | | | | | | | |
| Llama-3-8B | 94.65 ± 1.23 | 92.60 ± 0.12 | 97.62 ± 0.03 | 98.09 ± 0.10 | 97.74 ± 0.03 | 97.28 ± 0.08 | 88.62 ± 0.26 | 99.06 ± 0.05 | 99.05 ± 0.01 |
| **Strong GCN4LP baseline** | | | | | | | | | |
| NCN | 86.65 ± 5.37 | 96.66 ± 1.14 | 97.30 ± 0.26 | 98.66 ± 0.18 | 98.46 ± 0.19 | 97.77 ± 0.30 | 96.58 ± 0.2 | 98.96 ± 0.07 | 98.18 ± 0.20 |
| NCNC | 86.87 ± 7.99 | 96.56 ± 1.04 | 97.42 ± 0.37 | 98.66 ± 0.12 | 98.45 ± 0.21 | 97.79 ± 0.25 | 96.79 ± 0.25 | 98.93 ± 0.13 | 98.68 ± 0.06 |
| **FT-PLM-MLP** | | | | | | | | | |
| mpnet-FT | 85.93 ± 5.86 | 94.71 ± 1.16 | 97.36 ± 0.33 | 98.06 ± 0.19 | 97.72 ± 0.54 | 93.91 ± 0.64 | 82.26 ± 0.92 | 98.21 ± 0.26 | 98.17 ± 0.81 |
| e5-large-v2-FT | 86.95 ± 4.93 | 94.27 ± 0.85 | 97.39 ± 0.33 | 97.64 ± 0.36 | 94.06 ± 0.84 | 94.82 ± 1.18 | 86.26 ± 1.16 | 97.68 ± 0.17 | 97.08 ± 1.42 |
| MiniLM-L6-v2-FT | 87.09 ± 2.51 | 93.98 ± 0.85 | 97.25 ± 0.36 | 97.79 ± 0.14 | 97.95 ± 0.44 | 95.58 ± 0.51 | 88.34 ± 0.75 | 98.80 ± 0.16 | 97.85 ± 1.83 |
| **PLM-Inf-GCN** | | | | | | | | | |
| MiniLM-NCN | 87.15 ± 6.84 | 96.93 ± 0.54 | 86.69 ± 0.28 | 98.97 ± 0.10 | 98.99 ± 0.16 | 99.42 ± 0.11 | 99.59 ± 0.03 | 99.58 ± 0.07 | 98.17 ± 0.42 |
| e5-large-NCN | 88.31 ± 4.99 | 96.72 ± 0.67 | 97.82 ± 0.22 | 97.24 ± 0.20 | 99.03 ± 0.18 | 99.50 ± 0.13 | 99.56 ± 0.04 | 99.52 ± 0.07 | 98.15 ± 0.42 |
| **Ours** | | | | | | | | | |
| MiniLM-LMGJOINT | 88.20 ± 5.93 | 97.79 ± 0.66 | 98.22 ± 0.30 | 98.30 ± 0.51 | 99.00 ± 0.15 | 99.14 + 0.02 | 99.58 ± 0.04 | 99.60 ± 0.07 | 99.54 ± 0.14 |
| mpnet-LMGJOINT | 89.36 ± 5.37 | 98.78 ± 1.02 | 98.79 ± 0.49 | 99.34 ± 0.22 | 99.34 ± 0.09 | 99.54 ± 0.01 | 99.63 ± 0.02 | 99.72 ± 0.04 | 99.76 ± 0.09 |

**Does GCN4LP Maintain Its Superiority?** Currently promising GCN4LP methods—such as SEAL, BUDDY and NeoGNN, do **NOT** show a significant advantage over GCNs in this setting. Nonetheless, NCN/NCNC maintains superior performance across both Hits@k and MRR metrics, establishing itself as the strongest baseline with optimal computational complexity, as shown in Figure 3. Overall, all aggregated methods, including both GCN and GCN4LP, consistently outperform the structure-only methods, reaffirming the effectiveness of GCN-based approaches as a robust foundational framework. In PLM-Inf-GCN category, we observed that NCNC's performance could be improved simply by replacing the original node feature with PLM-based text embeddings. This signifies the significant potential of PLM-based text embeddings to enhance performance in link prediction tasks.

**Cascade PLM-GCN vs. Nested PLM-GCN.** To answer Q3, we evaluate the impact of fine-tuning within cascade and nested frameworks by comparing NCNC and PLM-Inf-NCNC, PLM-Inf-MLP and FT-PLM-MLP, GCN and FT-PLM-GCN. We observe limited improvement in Hits@K, accompanied by a notable decline in MRR. It indicates that incorporating context-aware PLM text encoding can enhance representation quality concerning Hits@K. However, the performance decay in MRR is consistent with prior findings by Chen et al. (2024b), which suggest that fine-tuning without specific design considerations can sometimes result in negative performance gains. In summary, these results indicate that optimizing PLMs independently of topology-based methods does not lead to consistent improvements across all metrics. This underscores the importance of developing an effective nested architecture to fully leverage the strengths of both approaches.

## 6.1 EMPIRICAL EVALUATION OF THE PROPOSED METHOD

We evaluate LMGJOINT through two complementary studies. First, a horizontal analysis in Experiment 1 (Exp) compares LMGJOINT against diverse graph-based methods on three popular datasets (Table 2). Second, a vertical analysis in Exp 2 examines competitive baselines in Exp 1 and broader LM-based approaches, extending the evaluation to nine datasets (Table 3) to assess the generality and robustness of LMGJOINT's improvements.

**Exp 1: Horizontal perspective** From Table 2, LMGJOINT consistently demonstrates superior performance across the three datasets and extensive metrics, outperforming the second-best category, PLM-Inf-GCN and strong baselines such as NCNC and LLaMA3-inf-MLP. It achieves the best results in 7 out of 9 comparative evaluations, highlighting its robustness and effectiveness. Compared to all benchmarked models, PLM/LLM-Inf-GCN family excels in Hits@K predominantly, while those with the highest MRR scores are concentrated within the Structure and GCN4LP categories. In contrast, our approach excels in both Hits@K and MRR, indicating its ability to preserve both the feature proximity from sentence embeddings and the structural proximity from graph topology.

**Exp 2 : Vertical perspective** In Table 3, we selected powerful baselines from Exp 1, including NCN(C), FT-PLM and PLM/LLM-Inf-NCN and extend to all proposed graphs (Except ogbn-paper100M due to GPU limitation). When comparing LM/LLM from a vertical perspective, Word2Vec outperforms the bag-of-words method among non-contextual embeddings, while local sentence embeddings improve AUC performance by over 10% compared to non-contextual embeddings in AUC. We observe a strong positive correlation between model performance gains and the number of parameters in PLM. LLaMA3 leads among LMs, achieving the best performance on pwc_small, indicating that text-only methods with large PLMs excel when structural information is limited. Nevertheless, Our LMGJOINT consistently outperforms these baselines across all evaluated datasets, achieving up to a 2.6% improvement in AUC.

## 7 CONCLUSION

This work tackles the under-explored area of joint PLM-GCN architecture design for link prediction by benchmarking PLM and GCN-based methods on extensive datasets. In our benchmark, we focused on discussing the impact of fine-tuning modules and text embeddings within various PLM-GCN architectures. We introduce the LMGJOINT, a simple yet powerful nested framework that combines the strengths of both GCN4LP and PLM-based methods while avoiding their weaknesses. Extensive and rigorous experiments demonstrate that LMGJOINT consistently improves performance across all the metrics on various datasets, with minimal hyperparameter tuning required. We expect it to benefit PLM-GCN models and other graph learning tasks, including node classification and regression.

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
