# A   THEORETICAL ANALYSIS

Given an undirected unweighted graph $\mathcal{G} = (\mathcal{V}, \mathcal{E}, \mathbf{A}, \mathbf{T})$, the inference problem is to approximate a conditional probability defined as Eq. 6.

$$\theta = \arg\max_{\theta_{\mathcal{G}, \mathbf{T}}} P(\mathbf{Y} \mid \mathbf{T}, \mathbf{A}) \tag{6}$$

Let $\mathbf{T}$ and $\mathbf{A}$ represent textual and topological features respectively, which are statistically dependent. $\theta_{\mathcal{G}, \mathbf{T}}$ refers to the parameters in GCN and LLM to be optimized.

**Theorem A.1** *Given a markov data processing pipeline $T \rightarrow (X, A) \rightarrow H \rightarrow Y$, we show that a residual connection from text embedding $X$ to the final prediction $Y$ increase the mutual information $I(T; Y)$.*

Proof of the Theorem A.1 Proof. To simplify the objective without compromising the conceptual integrity, we rewrite Eq. 6 using notion of mutual information. Next, we want to find a formula that connects the information from $Y$ and $T$ before and after the residual connection.

To begin, given the data processing markov chain $T \rightarrow (X, A) \rightarrow H \rightarrow Y$, where $A$ is statistically dependent from $T$. Using chain rule of mutual information, we have:

$$I(T; Y) = I(T; H) + I(T; Y \mid H) \tag{7}$$

Since $T$ and $Y$ are conditionally independent given $H$, we have:

$$I(T; Y \mid H) = 0 \tag{8}$$

Thus:

$$I(T; Y) = I(T; H) \tag{9}$$

In second case, with residual connection from $X \rightarrow Y$

$$I'(T; Y) = I(T; H, Y) \tag{10}$$
$$= I(T; H) + I(T; Y \mid H) \tag{11}$$
$$= I(T; H) + I(T; Y, X \mid H) - I(T; X \mid H) \tag{12}$$
$$= I(T; H) + I(T; X \mid H) + I(T; Y \mid X, H) - I(T; X \mid H) \tag{13}$$

For $I'(T; Y) > I(T; Y)$, the following condition must hold:

$$I(T; Y \mid X, H) > 0 \tag{14}$$

which means that when $X$ gives you additional predictive power about $Y$ beyond what you would know just from $H$.

**Theorem A.2** *Given positive and negative samples follow two Gaussian distributions with the same variance, we show that estimated Hits@1 and MRR are equivalent.*

Proof of the Theorem A.2

Proof. To simplify the objective without compromising the conceptual integrity, we rewrite two metrics using the Gaussian distribution notion. Next, we want to find a formula connecting their derived forms.

Positive samples: Let the distribution be $\mathcal{N}(-m, \sigma^2)$, where $m$ is the mean and $\sigma^2$ is the variance. Negative samples: Let the distribution be $\mathcal{N}(m, \sigma^2)$, where $m$ is the mean and $\sigma^2$ is the variance.

We further unfold two Gaussian distribution's probability density functions (pdfs): The PDF of the positive samples is:

$$f_{\text{pos}}(x) = \frac{1}{\sqrt{2\pi\sigma^2}} \exp\left(-\frac{(x+m)^2}{2\sigma^2}\right) \tag{15}$$

The PDF of the negative samples is:

$$f_{\text{neg}}(x) = \frac{1}{\sqrt{2\pi\sigma^2}} \exp\left(-\frac{(x-m)^2}{2\sigma^2}\right) \tag{16}$$

**Estimating Hits@1** It can be interpreted as the probability that a randomly chosen sample from one distribution will have a higher score than a randomly chosen sample from the other distribution. i.e. we need to find:

$$P(X_{\text{neg}} > X_{\text{pos}}) \tag{17}$$

We derive the above equation in the form of error function:

$$D = X_{\text{neg}} - X_{\text{pos}} \tag{18}$$

Then $D$ is also normally distributed because it is a linear combination of two normal variables:

$$D \sim \mathcal{N}(2m, 2\sigma^2) \tag{19}$$

The mean of $D$ is $2m$, and the variance of $D$ is $2\sigma^2$.

The Hits@1 is essentially the probability that the difference $D$ is greater than 0:

$$P(D > 0) = P\left(\frac{D - 2m}{\sqrt{2\sigma^2}} > \frac{0 - 2m}{\sqrt{2\sigma^2}}\right) \tag{20}$$

This can be rewritten in terms of the standard normal distribution $\Phi(\cdot)$:

$$P(D > 0) = \Phi\left(\frac{2m}{\sqrt{2\sigma^2}}\right) = \Phi\left(\frac{m}{\sigma}\sqrt{2}\right) \tag{21}$$

where $\Phi(\cdot)$ is the cumulative distribution function (CDF) of the standard normal distribution.

Therefore, Hits@1 can be estimated as:

$$\text{Hits@1} \approx \Phi\left(\frac{m}{\sigma}\sqrt{2}\right) \tag{22}$$

**Estimate MRR in Generalized Form** To estimate the value of the MRR given the means and variance of the two Gaussian distributions for positive and negative samples, The MRR is calculated as:

$$\text{MRR} = \frac{1}{N}\sum_{i=1}^{N}\frac{1}{\text{rank}_i} \tag{23}$$

where $\text{rank}_i$ is the position of the correct answer for the $i$-th query.

The probability that a positive sample has rank $r$ can be approximated by the probability that it is ranked higher than exactly $r - 1$ negative samples:

$$P(\text{rank} = r) = \binom{N_{\text{neg}}}{r-1}p^{r-1}(1-p)^{N_{\text{neg}}-(r-1)} \tag{24}$$

To align with the notion in Hits@1, we approximate the rank distribution using a Gaussian approximation to the binomial distribution.

Using the normal approximation to the binomial distribution:

$$P(\text{rank} = r) \approx \frac{1}{\sqrt{2\pi\sigma^2}}\exp\left(-\frac{(r - 1 - N_{\text{neg}}p)^2}{2\sigma^2}\right), \tag{25}$$

where $\sigma^2 = N_{\text{neg}}p(1-p)$.

The MRR is the expected value of the reciprocal of the rank. Therefore, we can rewrite MRR by substituting the Gaussian approximation for $P(\text{rank} = r)$ as:

$$\text{MRR} \approx \sum_{r=1}^{N_{\text{neg}}}\frac{1}{r}\frac{1}{\sqrt{2\pi N_{\text{neg}}p(1-p)}}\exp\left(-\frac{(r - 1 - N_{\text{neg}}p)^2}{2N_{\text{neg}}p(1-p)}\right). \tag{26}$$

For large $N_{\text{neg}}$, we can approximate the sum as an integral:

$$\text{MRR} \approx \int_{1}^{N_{\text{neg}}}\frac{1}{r}\frac{1}{\sqrt{2\pi N_{\text{neg}}p(1-p)}}\exp\left(-\frac{(r - 1 - N_{\text{neg}}p)^2}{2N_{\text{neg}}p(1-p)}\right)dr. \tag{27}$$

Define:

$$z = \frac{r - 1 - N_{\text{neg}}p}{\sqrt{N_{\text{neg}}p(1-p)}}. \tag{28}$$

Then, the integral can be further simplified:

$$\text{MRR} \approx \int_{-\frac{N_{\text{neg}}p+1}{\sqrt{N_{\text{neg}}p(1-p)}}}^{\frac{N_{\text{neg}}(1-p)-1}{\sqrt{N_{\text{neg}}p(1-p)}}} \frac{1}{(z\sqrt{N_{\text{neg}}p(1-p)} + N_{\text{neg}}p + 1)} \frac{1}{\sqrt{2\pi}} e^{-\frac{z^2}{2}} dz. \tag{29}$$

The integral represents the expectation of the function $\frac{1}{z\sqrt{N_{\text{neg}}p(1-p)}+N_{\text{neg}}p+1}$ under the standard normal distribution $e^{-\frac{z^2}{2}}$.

For large $N_{\text{neg}}$, the term $z\sqrt{N_{\text{neg}}p(1-p)}$ can be neglected comparing to $N_{\text{neg}}p$, so we approximate the denominator as:

$$\approx N_{\text{neg}}p. \tag{30}$$

Thus, the MRR approximately simplifies to:

$$\text{MRR} \approx \frac{1}{N_{\text{neg}}p} \int_{-\infty}^{\infty} \frac{1}{\sqrt{2\pi}} e^{-\frac{z^2}{2}} dz = \frac{1}{N_{\text{neg}}p}. \tag{31}$$

Therefore, the approximate value of MRR is:

$$\text{MRR} \approx \frac{1}{N_{\text{neg}}\Phi\left(\frac{m}{\sigma}\sqrt{2}\right)}, \tag{32}$$

This provides a reasonable approximation of MRR given the Gaussian-distributed ranks for large $N_{\text{neg}}$. Thus we proved Theorem A.2

**Theorem A.3** *For any node pair $(i, j)$, the approximation error of $[\mathbf{H}_C; \mathbf{H}_T] - \mathbf{H}_C + \mathbf{H}_T$ decreases as the number of nodes increases. We this by analyzing the distribution of the two random variables $\mathbf{H}_C$ and $\mathbf{H}_T$, and show that they have very limited overlapping.*

Proof of the Theorem A.3: Given a graph with $N$ nodes, and consider arbitrary pairs of nodes $(i, j)$. **According to (Mao et al., 2023) Lemma 2 (Incompatibility between LSP and FP factors).** For any $\delta > 0$, with probability at least $1 - 2\delta$, we have:

$$\eta_{ij} = \frac{c'}{1 - \beta_{ij}} + N(1 + \epsilon), \quad c' < 0 \tag{33}$$

$$\rightarrow \eta_{ij} - N + \frac{c}{1 - \beta_{ij}} = N\epsilon, \quad c > 0 \tag{34}$$

$$\rightarrow \frac{1}{N}\left(\eta_{ij} - N + \frac{c}{1 - \beta_{ij}}\right) = \epsilon \tag{35}$$

where $\eta_{ij}$ and $\beta_{ij}$ are the number of common neighbor nodes and feature proximity between nodes $i$ and $j$, respectively. We normalize $\eta_{ij}$ by dividing both sides by $N$, $c' < 0, c > 0, c = -c'$ is an independent variable that does not change with $\beta_{ij}$ and $\eta_{ij}$.

We derive this lemma in the notion of $\mathbf{H}_C$ and $\mathbf{H}_T$ first. We rewrite Lemma 1 as following: For any $\delta > 0$, with probability at least $1 - 2\delta$, we have:

$$P\left[\left|\frac{1}{N}\left(\eta_{ij} - N + \frac{c}{1 - \beta_{ij}}\right)\right| < \epsilon\right] > 1 - 2\sigma \tag{36}$$

$$\rightarrow \lim_{N\to\infty} P\left[\left|\frac{1}{N}\left(\eta_{ij} - N + \frac{c}{1 - \beta_{ij}}\right)\right| < \epsilon\right] = 1 \tag{37}$$

$$\rightarrow \lim_{N\to\infty} \frac{1}{N}\left(\eta_{ij} - N + \frac{c}{1 - \beta_{ij}}\right) = 0 \tag{38}$$

$$\rightarrow \lim_{N\to\infty} \left(\eta_{ij} - N + \frac{c}{1 - \beta_{ij}}\right) = 0 \tag{39}$$

$$\rightarrow \eta_{ij} - N + \frac{c}{1 - \beta_{ij}} = 0 \tag{40}$$

Then, let's analyse the distribution of $\eta_{ij}$ and $\beta_{ij}$. We derive the distribution of $\eta_{ij}$ using the Lemma 11.9 in Brede (2012). For each node, there are $\binom{n-1}{2}$ pairs of others with which it could form a common neighbor, and each common neighbor is present with probability $\frac{c}{\binom{n-1}{2}}$, for an average of $c$ common neighbors per node. Each common neighbor contributes two edges to the degree, so the average degree is $2c$. The probability $p_t$ of having $t$ common neighbors follows the binomial distribution:

$$p_t = \binom{\binom{n-1}{2}}{t} p^t (1-p)^{\binom{n-1}{2}-t} \approx e^{-c}\frac{c^t}{t!},$$

where $c$ is the mean degree, the final equality is exact in the limit of large $n$. The original probability distribution is given by:

$$p_t \approx \frac{e^{-c}c^t}{t!}$$

Using Stirling's approximation for large $t$, we have:

$$t! \approx \sqrt{2\pi t}\left(\frac{t}{e}\right)^t$$

Substituting Stirling's approximation into the original distribution:

$$p_t \approx \frac{e^{-c}c^t}{\sqrt{2\pi t}\left(\frac{t}{e}\right)^t}$$

Simplifying further, we obtain:

$$p_t \approx \frac{e^{t-c}c^t}{\sqrt{2\pi t}t^t}$$

We aim to find the distribution of $\beta_{ij} = 1 - \frac{c}{\tilde{\eta}_{ij}}$. In terms of $\alpha$, we can write this as:

$$\tilde{\eta}_{ij} = \frac{c}{1 - \beta_{ij}}$$

To simplify the derivation we write $\tilde{\eta}_{ij}$ as $\alpha$ and $\beta_{ij}$ as $\beta$. The chain rule of probability for transformations states that, if $X$ is a random variable with a known distribution, and we have a transformation $Y = g(X)$, then the probability distribution of $Y$ can be found as:

$$p_Y(y) = p_X(x)\left|\frac{dx}{dy}\right|,$$

where $\frac{dx}{dy}$ is the derivative of the inverse transformation.

$$\beta = 1 - \frac{c}{N - \alpha} \quad \Rightarrow \quad \alpha = N - \frac{c}{1 - \beta}.$$

The derivative of $\alpha$ with respect to $\beta$ is:

$$\frac{d\alpha}{d\beta} = \frac{c}{(1 - \beta)^2}.$$

Substituting the Poisson distribution for $p(\alpha)$:

$$p(\alpha = t) = e^{-c}\frac{c^t}{t!},$$

where $t = \frac{c}{1-\beta}$. So, for each value of $\beta$, the probability is:

$$p(\beta) = e^{-c}\frac{c^{\frac{c}{1-\beta}}}{\left(\frac{c}{1-\beta}\right)!} \cdot \frac{c}{(1-\beta)^2}.$$

The final probability distribution for $\beta$ after applying the chain rule can be written as:

$$p(\beta) = e^{-c}\frac{c^{\frac{c}{1-\beta}}}{\left(\frac{c}{1-\beta}\right)!} \cdot \frac{c}{(1-\beta)^2}.$$

Then we discuss about the behaviors of these two distributions. $\alpha$ follows a Poisson distribution where the mean and variance are both $\frac{2}{a(N-2)}$. $a$ is the mean degree. Let's analyze the behavior of $\beta$ in the range $\beta \in [0, 1)$ for the distribution:

$$p(\beta) \approx \frac{e^{-c+\frac{c}{1-\beta}}}{\sqrt{2\pi\frac{c}{1-\beta}}} \cdot \frac{c}{(1-\beta)^2}.$$

We'll explore its behavior across the full range $\beta \in [0, 1)$. The distribution consists of three primary terms: Exponential term: $e^{-c+\frac{c}{1-\beta}}$, Square root term: $\frac{1}{\sqrt{2\pi\frac{c}{1-\beta}}}$, Rational term: $\frac{c}{(1-\beta)^2}$. When $\beta \to 0$, we have $1 - \beta \approx 1$. Therefore, the terms simplify as:

$$p(0) \approx \frac{1}{\sqrt{2\pi c}} \cdot c = \frac{c}{\sqrt{2\pi c}} = \frac{\sqrt{c}}{\sqrt{2\pi}}.$$

When $\beta \to 1$, $1 - \beta$ becomes very small, and each term behaves as follows:

$$p(1) = 0.$$

For intermediate values of $\beta$, the behavior depends on how the three terms interact. The rational term $\frac{c}{(1-\beta)^2}$ increases as $\beta$ approaches 1, while the exponential term $e^{\frac{c}{1-\beta}}$ grows very rapidly for large $\beta$. Together, these terms initially cause an increase in $p(\beta)$, but the exponential decay from $e^{-c}$ causes the distribution to eventually decrease and approach 0 as $\beta \to 1$. Given the behavior of the distribution over $\beta \in [0, 1)$, we can approximate it in two regimes: Near $\beta = 0$, the distribution behaves approximately like:

$$p(\beta) \approx \frac{\sqrt{c}}{\sqrt{2\pi}}.$$

This is a constant value for small $\beta$. Near $\beta = 1$: The distribution decreases rapidly due to the exponential term. For large $\beta$, we can approximate:

$$p(\beta) \approx \frac{c}{(1-\beta)^2}e^{-\frac{c}{1-\beta}}.$$

This approximation shows that the distribution approaches 0 as $\beta$ nears 1. This analysis shows that the distribution is **monotonically decreasing** as $\beta$ approaches 1, after an initial phase where it remains approximately constant for small $\beta$. Thus, the distribution between these two variables has limited overlap.

## B  EXPERIMENT SETTING & HYPER PARAMETER RANGE

### B.1  TRAINING SETTING

In all Graph-agnostic, LLM-agnostic and proposed approach, we leverage binary cross entropy defined as:

$$L(u) = -\sum_{(i,j)\in\mathcal{E}^+}\log\sigma(\mathbf{h}_u, \mathbf{h}_v) - \sum_{(i,j)\in\mathcal{E}^-}\log(1-\sigma(\mathbf{h}_i, \mathbf{h}_j)).$$

where $\mathbf{h}_i$ and $\mathbf{h}_j$ are the node embeddings; $\langle \cdot, \cdot \rangle$ denotes the computation of the inner product; $\mathcal{E}_{\text{neg}}$ stands for the negative samples, where $\sigma$ is the sigmoid function, and $\mathcal{E}$ is the set of observed edges in the graph. The loss is optimized using the Adam optimizer Kingma & Ba (2014). During training we randomly sample one negative sample per positive sample. For GCN category, all models are trained for a maximum of 2000 epochs.

## B.2 EXPERIMENT AND TUNING DETAILS

We present the hyperparameter searching range in Table 2. and Table 3.

For the smaller graphs, Cora, Pubmed and Arxiv_2023, we utilize the following hyper-parameter search space. However, it's not feasible to tune over such large space for larger datasets. We utilize the average value of each optimized value from Cora, Pubmed, Arxiv_2023.

**GCN (Kipf & Welling, 2016), GAT (Velickovic et al., 2017), GraphSAGE Hamilton et al. (2017)**: learning rate: [0.01, 0.001, 0.0001], batch size: [$2^7$, $2^8$, $2^9$, $2^{10}$], dropout for hidden layer: [0.1, 0.3], weight decay: [1e-4, 1e-6], number of gcn layers: [1, 2, 3], number of mlp layers: [1, 2, 3], number of hidden layers: [$2^6$, $2^7$, $2^8$], dimension of output layer of gcn: [$2^6$, $2^7$, $2^8$]. Similar to (Chamberlain et al., 2023) we utilize a full adjacency matrix and remove the target link.

**NCNC (Wang et al., 2023)**: Our full list of hyperparameter for NCNC are: feature dropout: [0.0, 0.3, 0.7], lr for gnn: [0.001, 0.0001], lr for predictor: [0.001, 0.0001], hiddim dim: [64, 256], dropout for gnn: [0.0, 0.2, 0.5], dropout for dp: [0.0, 0.05], gnnlr: [0.001, 0.0001], prelr: [0.001, 0.0001], batch_size: [$2^7$, $2^8$, $2^9$, $2^{10}$]. We transferred the experiment setting from Wang et al. (2023) into our benchmark, including removing the validation edge during training, rest parameters, and JK connection.

**NeoGNN (Yun et al., 2021)**: hidden channels: [64, 128, 256, 512, 1024, 2048, 4096, 8192], num of layers: [1, 2, 3, 4], number of mlp layers: [1, 2, 3, 4], dropout': [0, 0.1, 0.2, 0.3, 0.4, 0.5], batch size: [128, 256, 512, 1024], learning rate: [0.01, 0.001, 0.0001]. We transferred the experiment setting from the original paper.

**SEAL (Zhang et al., 2020)**: hidden channels: [32, 64, 128, 256], batch size: [32, 64, 128, 256], lr: [0.001, 0.0001]. We transferred the experiment setting from the original paper, including text, structure embedding and

**BUDDY**: hidden channels: [128, 256, 512], batch size: [512, 1024], learning rate: [0.01, 0.001, 0.0001], maximum of hash hops: [1, 2, 3], dropout for label: [0.1, 0.5], dropout for feature: [0.1, 0.3, 0.5], dropout of sign: [0.1, 0.3, 0.5, 0.7]. We transferred the experiment setting from the original paper, including text, structure embedding, and utilizing valid edge during training.

**HLGNN Zhang et al. (2024b)**: hidden channels: [128, 256, 512, 1024, 2048, 4096, 8192], batch size: [128, 256, 512, 1024], learning rate: [0.01, 0.001, 0.0001], Initial parameter: [0.1, 0.2, 0.3, 0.4, 0.5], $\beta$ initialization: [RWR, KI], number of MLP layers: [2, 3, 4], dropout: [0.1, 0.2, 0.3, 0.4, 0.5, 0.6]. We transferred most experiment settings from the original paper, excluding the data split method and split ratio. We split the edges randomly while the original paper split it based on local structure. The original paper's split rate is 5%, 10% and 84% for test/valid/train edges. Ours is 5%, 15% and 80%.

**MiniLM (Reimers & Gurevych, 2019a)**: We leverage the [EOS] token in LLaMA3 and the [CLS] token in sentence embedding models as node features. We froze the layer up to the 6-th encoder layer in MiniLM (Reimers & Gurevych, 2019a); The utilized feature dimension, feat shrink, is set to 768. We use 'sentence-transformers/all-MiniLM-L6-v2' from Reimers & Gurevych (2019b). For training, the parameters include an attention dropout rate of 0.1, a batch size of 128, a classification dropout rate of 0.0, and a general dropout rate of 0.1. The model is trained for 250 epochs, with evaluation patience set to 1. Gradient accumulation steps are set to 1, and the learning rate is 0.0001. The warmup period is set to 0.6 epochs, and weight decay is set to 0.0.

**e5-large (Wang et al., 2022)**: We frozen up to 23-th encoder layer in e5-large and fine-tune rest layers. The utilized feature parameters dimension, feat shrink is 768. For training, the parameters include an attention dropout rate of 0.1, a batch size of 128, a classification dropout rate of 0.4, and a general dropout rate of 0.3. The model is trained for 250 epochs, with evaluation patience set to a

very large number (effectively disabling early stopping). Gradient accumulation steps are set to 1, the learning rate is 0.0001. The warmup period is set to 0.6 epochs, and weight decay is set to 0.0.

**MPNet** (Song et al., 2020): Same as Minilm, we use CLS token embedding, we frozen up to 32-th layer in Mpnet and fine-tune the rest layers. Feat shrink, is 768. The model name is 'sentence-transformers/all-mpnet-base-v2' from Reimers & Gurevych (2019b). For training, the parameters include an attention dropout rate of 0.1, a batch size of 256, a classification dropout rate of 0.4, and a general dropout rate of 0.3. The model is trained for 250 epochs, with evaluation patience set to a very large number, effectively disabling early stopping. Gradient accumulation steps are set to 4, and the learning rate is 0.0001. The warmup period is set to 0.6 epochs, and weight decay is set to 0.0.

**PLM-Inf-MLP**: We utilize BERT, MiniLM, e5-large and Llama 3 8B as inference models to only embed text without training.

**FT-PLM-MLP**: We use the above parameters to find these sentence transformers followed by an MLP.

**PLM-Inf-GCN**: This setting requires two separate steps. First, we embed the raw text with the above sentence transformers with given parameters and save the embedded text as original node features. In the second step, we load this embedded text as node features to aggregated features using different GCNs with the same setting as above.

**FT-PLM-GCN**: We leverage the same fine-tuned setting, the same parameters of sentence transformers and GCNs as above. We use the same split and full adjacency matrix during the training as category GCNs.

**LMGJOINT**: We utilize the same fine-tuned setting of Mpnet, Minilm, and e5-large in our proposed method without any additional parameter tuning, We leverage the same data split and full adjacency matrix for common neighbor and GCN, validation edge during training. In addition, the same parameters for optimizer and GCNs from (Kipf & Welling, 2016). The parameters for GCN are listed below: an input channel size of 1433, a hidden channel size of 256, and an output channel size of 32. It consists of 1 multi-layer perceptron (MLP) layer and 3 such graph convolution layers. We only tune the parameters for the optimizer including learning rate, batch size and weight decay. The weight of the soft common neighbor is tuned to be 0.1.

### B.3 LLM FINE-TUNE DETAILS

We leverage the [EOS] token in LLaMA3 and the [CLS] token in sentence embedding models as node features. We frozen the layer up to 6-th encoder layer in MiniLM (Reimers & Gurevych, 2019a); up to 32-th layer in Llama3 (Dubey et al., 2024); up to 12-th encoder layer in BERT and up to 23-th encoder layer in e5-large (Wang et al., 2022).

## C HARDWARE SPECIFICATION

We run experiments on our benchmarks with an Horeka Cluster, which features an 20-core CPU, 140 GB Memory and a Nvidia A100 GPU with 40 GB GPU Memory.

### C.1 COMPLEXITY ANALYSIS: DETAILS

**LLM Complexity Analysis** Transformer-based LLM's time complexity is $O(l(Nh^2 + N^2h))$, with space complexity $O(l \times H^2)$, where $N$, $h$, $l$, $D_k$ denotes sequence length, token hidden dimension, number of transformer layers, dimension of each attention head.

**GNN Complexity Analysis:** Given a graph $\mathcal{G} = (\mathbf{A}, \mathbf{X})$, the complexity of an $L_g$-layer GCN is $L_g \times (O(dhN) + O(whN))$, where $0 \leqslant d \leqslant 1$, and $d$, $h$, $N$, $w$ denote the average degree, dimension of node features, number of nodes, and dimension of hidden channels, respectively. For a large-scale graph as $N \to \infty$, and assuming the average in-degree is relatively small, the leading term is $O(L_g Nhw)$. In most experiments, we have seen a stabilizing performance achieving the peak within 3 layers. So we may choose to fix a number $L_g = 3$ of layers to perform. We rewrite $O(L_g Nhw)$ as $O(hNW)$, where $W$ serves as a composite indicator encapsulating the complexity of the GCN. Thus, the computation complexity for forward feature aggregations and backward gradient

aggregations is approximately $O(2hNW)$ per epoch. The complexity of the GCN is influenced by the previous LLM solely through the parameter $h$.

**LLM as static embedder:** Suppose $N$ is the total number of nodes and the average sequence length is $S$. The time complexity of encoding all graph features is $O(l(h^2NS + hN^2S^2))$, which increases quadratically with the graph size (number of nodes) but is independent of the graph density (number of edges).

**LLM-as-predictor:** In this scenario, the LLM's embeddings are directly connected to an MLP with $m$ layers and $p$ neurons. The total complexity is $O(l(h^2NS + hN^2S^2)) + O(N^2(2hp + mp^2))$. Since $m$ is typically small ($m \leq 3$) and the optimal range for $p$ is between $2^5$ and $2^7$, the expression can be conventionally simplified to $O(lhN^2S^2 + N^2mp^2)$, isolating the dominant term and ignoring constant factors.

**LLM-GCN nested architecture:** Similar to the LLM-as-predictor approach, we incorporate a GCN between the LLM and MLP as a subsequent structure embedder, resulting in a training complexity of $O(lhN^2S^2 + N^2mp^2 + 2hNW)$. As $N$ and $S$ approach infinity, we can disregard the less significant terms $O(2hNW + N^2mp^2)$, given that the GCN does not substantially contribute to computational overhead and the MLP can be effectively replaced by a dot product without appreciable loss in performance or increase in latency. This simplifies the complexity to $O(lhS^2N^2)$.

## D    CHOICES OF METRICS: DETAILS

The work Hu et al. (2021) has suggested different metric for each dataset. It including Mean Reciprocal Rank (MRR), Hits@K, and area under the curve (AUC) from recommendation system, knowledge graph completion and graph embedding respectively.

**MRR**, calculates the average reciprocal rank of the true positive among negative candidates. Yet, a significant challenge arises due to the fact that most datasets do not provide pre-defined negative samples, resulting in high computational costs for evaluation.

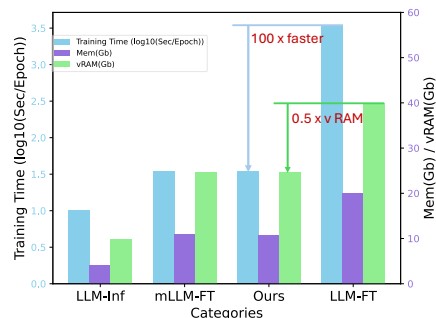

Figure 4: Comparison of training time (log-scale), memory usage and vRAM across different training categories. The left y-axis shows the training time in seconds per epoch (log scale), while the right y-axis displays memory usage and vRAM in gigabytes (Gb).

**AUC** is a scale-invarint metric, measuring the area beneath the FPR-TPR curve at various thresholds, a model's ability to distinguish between positive and negative edges, i.e., $p(\text{pos}) > p(\text{neg})$. However, as dominant algorithms have approached 95%, AUC's discriminative power has neared saturation. AUC gauges the likelihood that a positive sample is ranked higher than a randomly selected negative sample.

$$\text{AUC} = \frac{1}{|D_0| \cdot |D_1|} \sum_{i \in D_0} \sum_{j \in D_1} 1(R_i < R_j) \tag{41}$$

In this context, $D_0$ denotes the set of positive samples, $D_1$ is the set of negative samples, and $R_i$ represents the rank of the $i$-th sample. The indicator function $R_i < R_j$ equals 1 if $R_i < R_j$, and 0 otherwise.

**Hits@K**, quantifies the ratio of positive edges ranked within the top K positions. However, due to the lack of scale invariance, Hits@50 is usually reported for smaller datasets while Hits@100 for larger graphs. The choice of K significantly impacts model ranking, introducing a subtle yet important bias. While Hits@1 can help bridge this gap, due to the limited performance of current algorithms, it is highly sensitive to variance and hyper-parameters.

$$\text{Hits@k} = \frac{1}{N} \sum_{i=1}^{N} \mathbf{1}(\text{rank}_i \leq k) \tag{42}$$

Here, $R_i$ represents the rank of the $i$-the sample, and $\mathbb{I}(R_i \leq K)$ is an indicator function, taking the value 1 if $R_i \leq K$, and 0 otherwise.

**Generalized MRR** however, can be conveniently applied without aligning the source node, i.e. randomly sampling both positive and negative candidates. In this context, MRR evaluates the model's capacity to rank all positive samples above all negative ones, particularly distinguishing weak positives from strong negatives, rendering it one of the most stringent metrics available. Its performance is notably correlated with Hits@1, we leverage MRR in this setting to reduce evaluation cost and assess the performance ceiling of the most competitive models.

**Mean Reciprocal Rank (MRR):** This metric represents the mean of the reciprocal rank across all positive samples. Here, $R_i$ signifies the rank of the $i$-th sample.

$$\text{MRR} = \frac{1}{N} \sum_{i=1}^{N} \frac{1}{R_i}, \quad R_i = \frac{1}{N} \sum_{i=1}^{N} \frac{1}{\text{rank}_i} \tag{43}$$

For the datasets Cora, Citeseer, and Pubmed, due to the relatively small number of negative samples used in the evaluation (e.g., approximately 500 negatives for Cora and Citeseer), $K = 100$ is insufficient to differentiate between models such as GCN, GCN4LP, and LLM-related models. Therefore, we employed $Hits@50$ and MRR. When comparing the best models from LLM and GCN4LP, we use MRR.

We then prove that MRR in this setting is statistically linear dependent on Hits@1. See Section A.2

# E ERROR ANALYSIS

| Type I | Source Paper | Target Paper | Note |
|---|---|---|---|
| Ex 1 | Title: How Useful are **Educational** Questions Generated by Large Language Models? Abstract: Controllable text generation (CTG) by large language models has a huge potential to transform education for teachers and students alike. …The results demonstrate that the questions generated are high quality and sufficiently useful, showing their promise for widespread use in the classroom setting. … | Title: Opportunities and Risks of LLMs for Scalable Deliberation with Polis Abstract: Polis is a platform that leverages machine intelligence to scale up deliberative processes. In this paper, we explore the opportunities and risks associated with applying Large Language Models (LLMs) towards challenges with facilitating, moderating and summarizing the results of Polis engagements.…Finally, we conclude with several open future research directions for augmenting tools like Polis with LLMs. | |

| Type II | Source Paper | Target Paper | Note |
|---|---|---|---|
| Ex 1 | Title: Learning By Error-Driven decomposition Abstract: In this paper we describe a new self-organizing decomposition technique for learning high-dimensional mappings. Problem decomposition is performed in an error-driven manner, such that the resulting subtasks (patches) are equally well approximated. …The appropriateness of our general purpose method will be demonstrated with an example from mathematical function approximation. | Title: Exploration and Model Building in Mobile Robot Domains Abstract: I present first results on COLUMBUS, an autonomous mobile robot. COLUMBUS operates in initially unknown, structured environments. Its task is to explore and model the environment efficiently while avoiding collisions with obstacles. …COLUMBUS operates in real-time. It has been operating successfully in an office building environment for periods up to hours. | data noise |

Table 4: Examples of Type I (predict yes when no) and Type II (predict no when yes) returned from Llama. Correct keywords are written in **blue** and the captured feature words in the question are written in **bold**.

Given the promising performance of Llama 3, we conducted an error analysis to better understand its underlying mechanisms in link prediction. As shown in Table 4, we observed that Llama 3 tends to be sensitive to highly selective keywords and phrases but struggles to capture lexical variations and semantic relationships. We identify two types of errors: Type I and Type II. Type II errors are primarily caused by insufficient text; for example, without a connected database, mobile robot domains may be incorrectly predicted as irrelevant to error-driven decomposition. Additionally, papers cited in numerical methods, introductions and experiments may exhibit subtle or invisible textual and lexical relevance. In this context, structural features may play a critical role.

# F  LMGJOINT: DETAILS

## F.1  PSEUDOCODE & PIPELINE

In Fig. 1 we visualize LMGJOINT described in Section 4. We also give its pseudocode in Algorithm 1.

---

**Algorithm 1:** method: Framework for Link Prediction in Text Attributed Graph

**Input:** ( $\mathbf{A}, \mathbf{X}, \mathbf{T}, \mathcal{Y}, \mathcal{E}_{\text{train}}$ )
  Graph Adjacency Matrix $\mathbf{A} \in \{0,1\}^{n \times n}$, Init Node Feature $\mathbf{X}$;
  Node Feature In Raw Text $\mathbf{T}$;
  Set of positive and negative edge Labels $\mathcal{Y} \in \{1, 0\}$;
  Labeled positive and negative Edges $\mathcal{E}_{\text{train}}^+, \mathcal{E}_{\text{train}}^- \in \mathcal{E} = \{(v_i, v_j) \subseteq (\mathcal{V} \times \mathcal{V})\}$;
**Hyper-parameters:** Dropout Rate; Dimension of Feature Embedding $p$;
**Model Parameters:** PLM: $\mathbf{W}_{\mathcal{L}} \in \mathbb{R}^{F \times p}$; GCN: $\mathbf{W}_{\mathcal{G}} \in \mathbb{R}^{p \times |\mathcal{Y}|}$;
**Output:** Edge class label vector $\mathbf{Y}_{i,j}$

```
begin
    /* All new variables defined below are initialized as zero */

    /* Stage S1:  Text Embedding */
    for (i , j) ∈ 𝓔⁺_train ∪ 𝓔⁻_train do
        x_i^T ← σ(W_𝓛 t_i)
        x_j^T ← σ(W_𝓛 t_j) /* Textual Embeddings are stored in the matrix X^T */

        /* Stage S2:  Neighborhood Aggregation */
        /* Calculate Embedding for v_i and v_j and their Structural
           Proximity */

        x_i ← x_i^T
        x_j ← x_j^T      /* Replace x_i, x_j of current link with Textual Embedding
           x_i^T, x_j^T */
        for k ← 1 to K do
            /* symmetric degree-normalization of matrices S = D̃^(-½) Ã D̃^(-½) */
            H^𝒢 ← SXW_𝒢                     /* Final node embedding is H^K */

        H^K = f( Ã^K_sym X W_𝒢 )

        /* Stage S3:  Structural Feature Embedding */

        A ← dropout(A)
        /* Calculate Common Neighbor for v_i and v_j */

        H^C ← 𝕀[SUM(A_i ⊙ A_j) > d] > 0
        /* 𝕀 is a element-wise indicator function for matrix */

        /* Pairwise Text Feature Proximity in Dot Product */
        S^T = W(X_i^T ⊙ X_j^T)

    /* Structural and Textual Feature Concatenation */
    R ← (H^K ‖ S^T + βH^C)              /* β:  weight of Structural Proximity */
    R ← (R_-, R_+)
    /* Stage S4:  Binary Classification */
    R ← dropout(R)
    for (i , j) ∈ 𝓔_test do
        p_ij ← softmax(r_ij W);
        Y_{i,j} ← arg max(p_ij)                            /* edge label */
```

---

# G  PROPOSED DATASET: DETAILS

We provide the resource of collected datasets in Table 6 and systematic graph statistic in Table 5.

Table 5: Statistics of proposed dataset.

| | Pwc_small | Cora | PubMed | Arxiv_2023 | History | Photo | Pwc_medium | Ogbn_arxiv | Citationv8 | ogbn-papers100M |
|---|---|---|---|---|---|---|---|---|---|---|
| **#Nodes** $|\mathcal{V}|$ | 140 | 2708 | 19717 | 46198 | 36655 | 47420 | 86795 | 169343 | 1106759 | 111059956 |
| **#Edges** $|\mathcal{E}|$ | 798 | 10858 | 88648 | 77726 | 496248 | 872602 | 933411 | 2315598 | 12227452 | 1615685872 |
| **File Size (Mb)** | 0.139 | 2.57 | 31.01 | 57.26 | 57.12 | 37.44 | 99.58 | 190.27 | 1034.18 | 57344.12 |
| **#Split** | R | R | R | R | R | R | R | R | R | R |
| **Avg Deg (G)** | 8.51 | 11.63 | 19.27 | 44.38 | 47.26 | 108.95 | 614.64 | 330.02 | 74.28 | 68.23 |
| **Avg Deg (G2)** | 8.01 | 8.15 | 10.56 | 26.33 | 43.83 | 66.13 | 206.03 | 153.62 | 51.49 | 57.59 |
| **Clustering** | 0.22 | 0.24 | 0.06 | 0.13 | 0.28 | 0.38 | 0.13 | 0.20 | 0.13 | 0.18 |
| **Shortest Paths** | 3.47 | 6.31 | 6.34 | 6.06 | 5.38 | 4.76 | 4.23 | 5.58 | 5.92 | 6.02 |
| **Transitivity** | 0.22 | 0.09 | 0.05 | 0.04 | 0.17 | 0.12 | 0.01 | 0.02 | 0.03 | 0.01 |
| **Deg Gini** | 0.41 | 0.41 | 0.60 | 0.82 | 0.70 | 0.64 | 0.62 | 0.63 | 0.58 | 0.69 |
| **Coreness Gini** | 0.26 | 0.21 | 0.39 | 0.75 | 0.60 | 0.47 | 0.40 | 0.48 | 0.40 | 0.57 |
| **Heterogeneity** | -0.08 | 0.13 | 0.22 | 0.70 | 0.38 | 0.30 | 0.78 | 0.70 | 0.36 | 0.58 |
| **Power Law** $\alpha$ | 1.99 | 2.39 | 2.66 | 2.90 | 1.83 | 1.64 | 1.83 | 1.73 | 1.77 | 1.78 |
| **Edge Homophily** | 0.61 | 0.51 | 0.27 | 0.88 | 0.44 | 0.36 | 0.59 | 0.50 | 0.64 | 0.82 |
| **Jensen-Shannon** | 0.27 | 0.23 | 0.18 | 0.30 | 0.18 | 0.24 | 0.29 | 0.30 | 0.27 | 0.26 |
| **Mean Length** | 162 | 165 | 282 | 209 | 264 | 168 | 190 | 198 | 161 | 173 |

Table 6: M: million ($10^6$). The sampling rate for all datasets is at 5 minutes level.

| Released Source | Dataset | Nodes | Edges | Degree | Min-Max | Mean | Data Points |
|---|---|---|---|---|---|---|---|
| Ours | **PwC_small** | 140 | 798 | 8.51 | $79 - 293$ | 162 | 0.139M |
| Xiaoxin He et. al. He et al. (2023) | **Cora** | 2708 | 10858 | 11.63 | $8 - 929$ | 165 | 2.57M |
| Xiaoxin He et. al. He et al. (2023) | **Pubmed** | 19717 | 88648 | 19.27 | $4 - 994$ | 282 | 31.01M |
| Xiaoxin He et. al. He et al. (2023) | **Arxiv_2023** | 46198 | 77726 | 12.89 | $7 - 1617$ | 209 | 57.26M |
| Li, Zhuofeng et al. Shchur et al. (2018) | **Photo** | 47420 | 872602 | 108.95 | $4 - 7281$ | 168 | 37.44M |
| Li, Zhuofeng et al.(Li et al., 2024) | **History** | 36655 | 496248 | 47.26 | 42 - 329 | 264 | 42.16M |
| Ours | **PwC_medium** | 86795 | 933411 | 7.3 | $16 - 610$ | 190 | 99.58M |
| Weihua Hu et. al. Hu et al. (2021) | **Ogbn-Arxiv** | 169343 | 2315598 | 13.68 | $20 - 2214$ | 198 | 190.27M |
| Weihua Hu et. al. Wu et al. (2021) | **Citationv8** | 281142 | 938931 | 3.34 | $4 - 774$ | 161 | 1034.18M |
| Weihua Hu et. al. Hu et al. (2021) | **ogbn-paper100M** | 111059956 | 1615685872 | 13.68 | 42 - 213 | 173 | 537444.2M |

## G.1 CURRENT HYPOTHESIS CLASSES

In this section, we explore four commonly used homophily metrics derived from node classification: average connected feature similarity, generalized edge homophily Luan et al. (2023), and two distribution-based measurements, Hellinger Distance and Jensen-Shannon Divergence Luan et al. (2023). These metrics are defined as follows:

1. **Edge Homophily:** This metric measures the average similarity across all connected edges. It is defined as:

$$K = \sum_{(u,v)\in\mathcal{E}} \frac{k(u,v)}{|\mathcal{E}|} \quad \text{where } k(\mathbf{h}_i, \mathbf{h}_j) = \frac{\mathbf{h}_i \cdot \mathbf{h}_j}{\|\mathbf{h}_i\|\|\mathbf{h}_j\|}. \tag{44}$$

2. **Generalized Edge Homophily:** This approach employs normalized cosine similarity to compare the degree of similarity across all positive and negative samples within the test set. The accuracy is calculated as:

$$\text{Accuracy} = \frac{1}{N} \sum_{i=1}^{N} \mathbb{I}\left(q_i = \mathbb{T}(p_i)\right), \tag{45}$$

where $\mathbb{I}$ is the indicator function.

3. **Hellinger Distance:** This metric quantifies the similarity between two probability distributions, given the logits $\hat{\mathbf{P}}$ and true labels $\mathbf{Y}$:

$$H(P, Q) = \frac{1}{\sqrt{2}} \sqrt{\sum_i \left(\sqrt{P(i)} - \sqrt{Q(i)}\right)^2}. \tag{46}$$

4. **Jensen-Shannon Divergence:** This metric is used to measure the similarity between two probability distributions defined by the logits $\hat{\mathbf{P}}$ and true labels $\mathbf{Y}$:

$$D_{\text{JS}}(P \parallel Q) = \frac{1}{2} \sum_i P(i) \log \frac{P(i)}{M(i)} + \frac{1}{2} \sum_i Q(i) \log \frac{Q(i)}{M(i)}, \tag{47}$$

where $M(i) = \frac{P(i)+Q(i)}{2}$ represents the average distribution between $P(i)$ and $Q(i)$.

All these metrics are bounded within the range $[0, 1]$, with values approaching 1 indicating a high degree of edge homophily. These metrics serve as an initial exploration to uncover potential connections that can inform model design and dataset characteristics.

## G.2 STRUCTURE STATISTICS

Considerable work has demonstrated that local and global structural characterization's are more effective for LP. To translate the homophily assumption, local and global graph heuristics, small-world phenomenon, and scale-free network properties into task-specific statistics, we provided the following graph metrics. Text length and diversity are critical factors influencing the performance of a model. In this context, we present the distribution of text lengths within our dataset in Table 6.

1. **Graph Density**: Number of Nodes, Edges, Arithmetic Deg are used to measure the graph's size, density and sparsity. Average degree of each central node $v \in \mathcal{V}$ and its of 2-order neighborhood $\mathcal{N}_v$' average degree measures the graph's local connectivity.
2. **Graph Connectivity**: Clustering refers to fraction of possible triangles through one node exists:

$$C_i = \frac{2 \times T(i)}{\deg(i)(\deg(i) - 1)} \quad T = \frac{3 \times \text{\# triangles}}{\text{\# triads}} \tag{48}$$

where $T(i)$ is the number of triangles through node $deg(i)$ and $deg(i)$ is the degree of node $i$. Transitivity measures the fraction of all possible triangles in the graph.
3. **Hierarchical level**: We leverage k-Core graph's fraction and degree distribution to calculate Gini and Coreness Gini.
4. **Scale-free**: If its node degree distribution $P(d)$ follows a power law $P(d) \sim d^{-\gamma}$, where $\gamma$ typically lies within the range $2 < \gamma < 3$. We approximate power law $\alpha$ based on the following estimator. pwc_small, cora, pubmed, arxiv_2023, pwc_large are scale-free networks.

$$\hat{\alpha} = 1 + N \left( \sum_{i=1}^{n} \log \left( \frac{d_i + 1}{d_{\min} + 1} \right) \right)^{-1} \tag{49}$$

## G.3 DETAILED INFORMATION ABOUT DATASETS

In this work, we evaluate the performance of the models on the Link Prediction task using a diverse collection of reference datasets. Our analysis includes 10 datasets obtained from three main sources: Planetoid Sen et al. (2008a), Amazon Shchur et al. (2018) and OGB Hu et al. (2020a). The Planetoid datasets include *Cora* and *PubMed*. The Amazon datasets include *Photo*. The OGB datasets include *ogbn-arxiv*, *Citationv8*, *ogbn-papers100M*, and *Arxiv 2023*. And datasets without a group are *Pwc_small*, *Pwc_medium* and *History*. The detailed statistics for each dataset are presented in Table 5. Below are descriptions of each dataset that were used in the experiments:

- **Cora McCallum et al. (2000)**: It consists of 2,708 scientific publications, it contains 5,429 links and each paper is either cited or referenced by at least one other paper. Each publication in the dataset is described by a 0/1-value vector indicating the absence/presence of the corresponding word in the dictionary. The dictionary consists of 1,433 unique words.
- **Pubmed Sen et al. (2008b)**: It contains 19,717 scientific publications from the PubMed database about diabetes research. It includes a citation network with 44,338 links, where nodes represent publications and edges denote citation relationships. Each publication is characterized by a TF/IDF weighted word vector derived from a dictionary of 500 unique terms.

- **Photo Shchur et al. (2018)**: The Amazon Photos dataset represents a collaborative shopping network, where nodes correspond to one of eight product categories, and edges indicate co-purchase relationships between products. Each node is characterized by a fixed-size object vector with 745 dimensions, which captures the relevant attributes of the corresponding product.

- **Ogbn-arxiv Hu et al. (2020a)**: The ogbn-arxiv dataset is a directed graph representing a network of citations of computer science articles from arXiv indexed by MAG Wang et al. (2020). The nodes correspond to individual papers, each of which is described by a 128-dimensional vector of characteristics obtained from the embeddings of words in their titles and annotations, while the directed edges indicate the citation ratio.

- **Ogbn-papers100M Hu et al. (2020a)**: The ogbn-papers100M dataset is a large-scale directed citation graph containing 111 million articles indexed by MAG Wang et al. (2020), making it one of the largest node classification datasets available. Similar to ogbn-arxiv, each node represents a paper with characteristics derived from its title and annotation, while the edges indicate the citation ratio. Approximately 1.5 million of these nodes are arXiv documents covering up to 172 research areas.

- **Arxiv 2023 (He et al., 2023)**: The Arxiv 2023 dataset is a citation network in computer science from arXiv, in particular those published in 2023 or later. The nodes correspond to individual papers, and the edges represent citation relationships.

- **Pwc Medium (Saier et al., 2023)**: The Papers With Code (PWC) medium dataset is a comprehensive resource for exploring the use of research artifacts such as datasets, methods, models, and tasks. It includes rich meta-information, such as paper descriptions, categories, and links to the corresponding code and articles. In this work, we only utilize the title and abstract as node features.

- **Pwc Small**: In order to fill the dataset scale gap in a hundred nodes, we extracted one connected component from PWC medium dataset as a small graph.

- **History (Li et al., 2024)**: The Goodread-History is a dataset in book recommendations. The Goodreads datasets are the main source. Nodes represent meta information of nodes such as types of books and reviewers, while edges indicate book reviews. Node labels are assigned based on the book categories. The descriptions of books and user information are used as book and user node textual information. The task is to predict the preference of users for products.

- **Citationv8 (Yan et al., 2023)**: The CitationV8 dataset is a extracted graph extracted from DBLP (Tang et al., 2008). Node textual attributes in CitationV8 are derived from the titles and abstracts. Each edge represents a citation relationship between two papers.

## H  COMPLEXITY ANALYSIS

The overall complexity of a self-attention layer with multiple heads is: $\mathcal{O}(N^2 \cdot d_{\mathrm{model}} + N \cdot d_{\mathrm{model}} \cdot d_{\mathrm{ff}})$, where $N$, $H$ denotes token length (sequence length), the number of attention heads, $d_{\mathrm{model}}$ is the dimensionality of each key vector. $d_{\mathrm{tt}}$ is the hidden dimensionality of the feedforward network.

## I  BENCHMARK RESULT OF ALL MODELS

Table 7: Benchmark result of local and global heuristic, graph embedding, vanilla GCNs, GCN2LPs, LLM-Inf, LLM-FT.

| Models | Cora | | PubMed | | Arxiv 2023 | |
|---|---|---|---|---|---|---|
| | Hits@100 | AUC | Hits@100 | AUC | Hits@100 | AUC |
| CN | $50.36 \pm 0.03$ | $74.67 \pm 0.01$ | $33.32 \pm 0.02$ | $66.58 \pm 0.00$ | $27.20 \pm 0.01$ | $63.34 \pm 0.00$ |
| AA | $50.36 \pm 0.03$ | $74.83 \pm 0.01$ | $33.32 \pm 0.02$ | $66.59 \pm 0.00$ | $27.20 \pm 0.01$ | $63.36 \pm 0.00$ |
| RA | $50.36 \pm 0.03$ | $74.83 \pm 0.01$ | $33.32 \pm 0.02$ | $66.59 \pm 0.00$ | $27.20 \pm 0.01$ | $63.36 \pm 0.00$ |
| PPR/sym | $88.93 \pm 0.00$ | $87.72 \pm 0.01$ | $72.95 \pm 0.01$ | $78.19 \pm 0.01$ | $67.86 \pm 0.02$ | $78.90 \pm 0.01$ |
| Katz | $69.25 \pm 0.02$ | $82.95 \pm 0.01$ | $66.02 \pm 0.02$ | $82.52 \pm 0.00$ | $55.39 \pm 0.01$ | $76.84 \pm 0.00$ |
| DeepWalk | $86.99 \pm 0.05$ | $87.02 \pm 0.01$ | $72.30 \pm 0.01$ | $84.48 \pm 0.01$ | $22.13 \pm 0.10$ | $75.28 \pm 0.01$ |
| Node2Vec | $85.02 \pm 0.01$ | $85.89 \pm 0.02$ | $71.35 \pm 0.01$ | $80.60 \pm 0.04$ | $27.98 \pm 0.17$ | $76.76 \pm 0.02$ |
| GIN | $96.05 \pm 2.13$ | $94.00 \pm 1.38$ | $92.02 \pm 0.91$ | $97.79 \pm 0.16$ | $53.22 \pm 2.05$ | $78.93 \pm 0.49$ |
| GAT | $95.34 \pm 1.61$ | $93.77 \pm 0.97$ | $84.03 \pm 1.59$ | $96.90 \pm 0.39$ | $53.49 \pm 1.06$ | $82.86 \pm 1.02$ |
| SAGE | $95.10 \pm 1.57$ | $91.77 \pm 0.91$ | $93.36 \pm 0.70$ | $98.66 \pm 0.15$ | $56.69 \pm 3.13$ | $87.86 \pm 1.97$ |
| GCN | $96.20 \pm 1.71$ | $94.18 \pm 1.67$ | $89.59 \pm 1.73$ | $98.24 \pm 0.16$ | $52.46 \pm 1.44$ | $81.89 \pm 0.51$ |
| RotatE | $56.60 \pm 6.29$ | $61.57 \pm 3.88$ | $38.58 \pm 2.59$ | $80.89 \pm 0.40$ | $34.57 \pm 0.84$ | $76.73 \pm 0.44$ |
| TransE | $57.00 \pm 7.06$ | $64.19 \pm 3.83$ | $50.04 \pm 2.63$ | $85.29 \pm 0.97$ | $30.22 \pm 1.96$ | $73.03 \pm 1.04$ |
| ComplEx | $73.59 \pm 0.90$ | $75.71 \pm 1.09$ | $53.87 \pm 3.09$ | $89.60 \pm 0.51$ | $32.17 \pm 2.38$ | $73.31 \pm 0.61$ |
| DistMult | $73.12 \pm 0.28$ | $76.53 \pm 0.80$ | $54.82 \pm 2.95$ | $89.91 \pm 0.53$ | $32.37 \pm 1.62$ | $73.25 \pm 0.39$ |
| SEAL | $92.03 \pm 2.96$ | $95.11 \pm 1.07$ | $89.52 \pm 1.27$ | $98.81 \pm 0.14$ | $67.34 \pm 3.74$ | $97.36 \pm 0.11$ |
| NeoGNN | $90.04 \pm 2.02$ | $91.02 \pm 1.08$ | $81.25 \pm 8.14$ | $94.69 \pm 6.85$ | $69.34 \pm 8.56$ | $86.76 \pm 3.50$ |
| ELPH | $94.91 \pm 2.17$ | $92.63 \pm 1.90$ | $74.62 \pm 1.64$ | $95.80 \pm 0.39$ | $66.95 \pm 3.62$ | $87.09 \pm 1.22$ |
| BUDDY | $95.42 \pm 2.26$ | $93.35 \pm 1.41$ | $89.25 \pm 2.27$ | $97.92 \pm 0.17$ | $60.49 \pm 0.94$ | $84.33 \pm 0.60$ |
| NCN | $98.74 \pm 0.96$ | $96.66 \pm 1.14$ | $93.21 \pm 1.10$ | $98.66 \pm 0.18$ | $88.83 \pm 1.43$ | $97.30 \pm 0.26$ |
| NCNC | $98.67 \pm 0.76$ | $96.56 \pm 1.04$ | $93.74 \pm 0.25$ | $98.66 \pm 0.12$ | $89.13 \pm 2.08$ | $97.42 \pm 0.37$ |
| BERT-Inf | $56.90 \pm 3.26$ | $65.09 \pm 1.41$ | $48.73 \pm 1.43$ | $88.96 \pm 0.31$ | $48.74 \pm 1.15$ | $86.37 \pm 0.27$ |
| e5-large-v2 | $83.10 \pm 0.80$ | $83.87 \pm 0.23$ | $82.59 \pm 0.26$ | $96.73 \pm 0.03$ | $84.09 \pm 0.24$ | $95.72 \pm 0.01$ |
| MiniLM-L6-v2 | $92.99 \pm 0.00$ | $91.22 \pm 0.04$ | $81.90 \pm 0.03$ | $96.20 \pm 0.00$ | $77.62 \pm 0.03$ | $95.22 \pm 0.00$ |
| Llama-3-8B | $95.64 \pm 0.41$ | $92.60 \pm 0.12$ | $89.01 \pm 0.53$ | $98.09 \pm 0.10$ | $89.91 \pm 0.19$ | $97.62 \pm 0.04$ |
| Llama-BUDDY | $97.47 \pm 1.44$ | $93.97 \pm 0.88$ | $94.60 \pm 0.98$ | $98.75 \pm 0.18$ | $84.93 \pm 1.68$ | $96.67 \pm 0.38$ |
| e5-large-BUDDY | $97.86 \pm 0.66$ | $94.03 \pm 0.77$ | $93.95 \pm 0.34$ | $98.72 \pm 0.06$ | $77.43 \pm 3.01$ | $95.27 \pm 0.31$ |
| BERT-BUDDY | $81.42 \pm 2.86$ | $79.62 \pm 2.44$ | $78.54 \pm 1.26$ | $96.25 \pm 0.17$ | $43.30 \pm 1.65$ | $77.93 \pm 0.62$ |
| MiniLM-BUDDY | $97.55 \pm 1.32$ | $94.74 \pm 0.99$ | $93.56 \pm 0.66$ | $98.68 \pm 0.07$ | $47.12 \pm 2.93$ | $77.59 \pm 0.35$ |
| Llama-NCN | $98.26 \pm 0.77$ | $96.23 \pm 0.28$ | $96.30 \pm 0.62$ | $99.11 \pm 0.07$ | $92.64 \pm 0.52$ | $98.24 \pm 0.17$ |
| e5-large-NCN | $98.81 \pm 0.62$ | $96.72 \pm 0.67$ | $96.33 \pm 0.70$ | $99.10 \pm 0.09$ | $90.79 \pm 1.38$ | $97.82 \pm 0.22$ |
| BERT-NCN | $82.93 \pm 3.65$ | $84.91 \pm 2.53$ | $81.09 \pm 2.46$ | $97.24 \pm 0.20$ | $66.20 \pm 0.62$ | $86.94 \pm 0.27$ |
| MiniLM-NCN | $98.58 \pm 0.60$ | $96.93 \pm 0.54$ | $95.49 \pm 0.66$ | $98.97 \pm 0.10$ | $66.47 \pm 0.84$ | $86.69 \pm 0.28$ |
| Llama-NCNC | $98.73 \pm 0.65$ | $95.63 \pm 0.77$ | $92.39 \pm 1.46$ | $98.54 \pm 0.18$ | $91.90 \pm 1.33$ | $97.97 \pm 0.36$ |
| e5-large-NCNC | $98.81 \pm 0.74$ | $96.31 \pm 0.68$ | $96.69 \pm 0.56$ | $99.14 \pm 0.10$ | $90.46 \pm 1.14$ | $97.72 \pm 0.16$ |
| BERT-NCNC | $84.11 \pm 2.70$ | $84.05 \pm 2.85$ | $82.48 \pm 1.43$ | $97.40 \pm 0.20$ | $68.50 \pm 3.49$ | $89.20 \pm 0.30$ |
| MiniLM-NCNC | $98.81 \pm 0.49$ | $96.82 \pm 0.62$ | $96.11 \pm 0.60$ | $99.07 \pm 0.07$ | $70.61 \pm 2.24$ | $88.42 \pm 0.72$ |
| Llama-NeoGNN | $83.00 \pm 8.24$ | $87.08 \pm 3.07$ | $69.42 \pm 3.09$ | $94.90 \pm 0.52$ | $63.67 \pm 8.29$ | $86.94 \pm 1.22$ |
| e5-large-NeoGNN | $82.06 \pm 4.54$ | $87.06 \pm 1.69$ | $71.13 \pm 2.19$ | $92.21 \pm 5.85$ | $69.09 \pm 9.97$ | $88.32 \pm 1.81$ |
| BERT-NeoGNN | $87.59 \pm 3.79$ | $87.77 \pm 2.30$ | $70.85 \pm 1.71$ | $95.08 \pm 0.38$ | $65.43 \pm 8.49$ | $86.22 \pm 1.93$ |
| MiniLM-NeoGNN | $87.59 \pm 6.28$ | $88.66 \pm 2.63$ | $70.62 \pm 1.71$ | $95.08 \pm 0.38$ | $68.60 \pm 11.86$ | $86.10 \pm 0.75$ |
| BERT-FT | $96.99 \pm 1.36$ | $92.88 \pm 0.99$ | $84.45 \pm 2.92$ | $97.32 \pm 0.44$ | $87.56 \pm 2.05$ | $96.98 \pm 0.31$ |
| e5-large-v2-FT | $96.92 \pm 1.35$ | $94.27 \pm 0.85$ | $87.23 \pm 1.60$ | $97.79 \pm 0.14$ | $89.35 \pm 1.33$ | $97.39 \pm 0.33$ |
| MiniLM-L6-v2-FT | $96.68 \pm 1.69$ | $93.98 \pm 0.85$ | $86.80 \pm 1.98$ | $97.64 \pm 0.36$ | $88.38 \pm 1.06$ | $97.25 \pm 0.36$ |
| all-mpnet-base-v2 | $97.78 \pm 0.66$ | $94.71 \pm 1.16$ | $90.69 \pm 2.49$ | $98.06 \pm 0.19$ | $91.44 \pm 0.75$ | $97.36 \pm 0.33$ |
| MiniLM-GAT-CT | $76.99 \pm 6.58$ | $76.09 \pm 4.18$ | $43.75 \pm 5.46$ | $90.11 \pm 1.81$ | $25.18 \pm 4.17$ | $80.31 \pm 3.08$ |
| MiniLM-SAGE-CT | $82.01 \pm 4.19$ | $80.15 \pm 2.74$ | $63.18 \pm 1.34$ | $94.10 \pm 0.21$ | $66.06 \pm 2.60$ | $92.63 \pm 0.51$ |
| mpnet-GIN-CT | $97.55 \pm 1.86$ | $92.30 \pm 2.28$ | $63.42 \pm 2.77$ | $94.82 \pm 0.42$ | $64.38 \pm 3.22$ | $91.97 \pm 1.07$ |
| mpnet-GAT-CT | $86.96 \pm 6.71$ | $86.64 \pm 5.61$ | $52.43 \pm 4.18$ | $91.75 \pm 0.86$ | $29.43 \pm 3.60$ | $82.50 \pm 3.28$ |
| mpnet-SAGE-CT | $93.88 \pm 0.28$ | $89.64 \pm 0.96$ | $71.62 \pm 2.78$ | $94.60 \pm 0.58$ | $66.28 \pm 4.50$ | $92.63 \pm 0.87$ |
| MiniLM-LMGJOINT-C | $99.92 \pm 0.18$ | $98.24 \pm 0.95$ | $99.94 \pm 0.08$ | $99.73 \pm 0.03$ | $98.16 \pm 1.73$ | $99.05 \pm 0.25$ |
| MiniLM-LMGJOINT | $99.84 \pm 0.35$ | $97.79 \pm 0.66$ | $89.22 \pm 4.96$ | $98.30 \pm 0.51$ | $91.44 \pm 1.34$ | $98.22 \pm 0.30$ |
| mpnet-LMGJOINT-C | $96.28 \pm 8.31$ | $94.68 \pm 6.81$ | $99.95 \pm 0.08$ | $99.42 \pm 0.38$ | $77.99 \pm 17.20$ | $92.88 \pm 6.53$ |
| mpnet-LMGJOINT | $100.00 \pm 0.00$ | $98.78 \pm 1.02$ | $97.13 \pm 1.74$ | $99.34 \pm 0.22$ | $94.85 \pm 3.15$ | $98.79 \pm 0.49$ |
| e5-large-LMGJOINT-C | $99.92 \pm 0.18$ | $98.36 \pm 1.15$ | $100.00 \pm 0.00$ | $99.49 \pm 0.18$ | $98.01 \pm 0.72$ | $98.51 \pm 0.12$ |
| e5-large-LMGJOINT | $98.89 \pm 1.02$ | $97.13 \pm 1.22$ | $88.41 \pm 1.14$ | $98.18 \pm 0.12$ | $87.71 \pm 1.48$ | $97.51 \pm 0.29$ |

Table 8: Benchmark result of local and global heuristic, graph embedding, vanilla GCNs, GCN2LPs, LLM-Inf, LLM-FT.

| Models | Cora | | PubMed | | Arxiv 2023 | |
|---|---|---|---|---|---|---|
| | Hits@50 | MRR | Hits@50 | MRR | Hits@50 | MRR |
| CN | 50.36±0.03 | 32.88±0.09 | 33.32±0.02 | 21.13±0.02 | 27.20±0.01 | 14.66±0.06 |
| AA | 50.36±0.03 | 47.33 ± 0.09 | 33.32±0.02 | 24.61±0.11 | 27.20±0.01 | 19.87±0.30 |
| RA | 50.36±0.03 | 47.17±0.11 | 33.32±0.02 | 23.94±0.16 | 27.20±0.01 | 19.16±0.27 |
| PPR/sym | 84.74±0.00 | 58.86±0.98 | 69.81±0.02 | 28.04±0.91 | 65.68±0.02 | 26.57±0.82 |
| Katz | 69.25±0.02 | 38.17±0.12 | 66.02±0.02 | 30.94±0.08 | 55.39±0.01 | 21.76±0.21 |
| DeepWalk | 81.07±0.02 | 33.88±0.81 | 64.97±0.03 | 25.49±0.52 | 13.57±0.08 | 2.54±0.00 |
| Node2Vec | 80.38±0.03 | 38.76±0.60 | 64.57±0.03 | 19.48±0.15 | 19.12±0.21 | 3.86±0.02 |
| GCN | 91.46±2.36 | 45.84±8.40 | 83.11±2.19 | 24.55±4.02 | 45.07±0.87 | 17.62±3.34 |
| GAT | 89.80±2.00 | 49.82±10.04 | 74.23±2.54 | 18.13±5.81 | 43.09±1.22 | 13.58±4.33 |
| SAGE | 86.40±3.73 | 46.03±6.70 | 86.55±0.55 | 35.63±5.75 | 45.42±3.12 | 11.52±1.67 |
| GIN | 91.54±2.91 | 51.90±6.65 | 86.92±1.68 | 24.63±2.24 | 45.35±2.58 | 14.79±4.53 |
| RotatE | 27.51 ± 6.39 | 2.98 ± 0.41 | 26.10 ± 2.89 | 4.41 ± 1.24 | 24.76 ± 1.31 | 5.65 ± 1.20 |
| TransE | 32.17 ± 6.52 | 3.97 ± 0.90 | 38.58 ± 3.64 | 7.74 ± 1.71 | 21.27 ± 1.81 | 5.11 ± 1.16 |
| ComplEx | 62.53 ± 1.71 | 17.82 ± 2.58 | 38.14 ± 3.55 | 6.50 ± 2.48 | 22.33 ± 1.05 | 5.16 ± 1.07 |
| DistMult | 64.90 ± 1.59 | 19.28 ± 2.11 | 39.09 ± 3.93 | 6.90 ± 2.50 | 21.32 ± 0.45 | 5.02 ± 1.24 |
| SEAL | 87.38±3.06 | 37.81±9.93 | 84.62±3.53 | 49.02±13.91 | 56.98±1.89 | 22.47±3.69 |
| NeoGNN | 81.03±3.11 | 41.48±5.11 | 73.17±5.29 | 31.44±3.85 | 64.54±11.14 | 28.07±15.62 |
| ELPH | 87.30±4.94 | 39.86±10.20 | 59.19±5.58 | 24.61±3.17 | 57.66±1.55 | 29.22±5.95 |
| BUDDY | 87.82±3.41 | 30.78±5.55 | 76.14±3.46 | 19.46±2.42 | 52.25±2.01 | 18.75±3.71 |
| NCN | 96.16±1.62 | 45.76±16.39 | 86.44±2.03 | 25.92±4.33 | 82.34±2.45 | 37.92±13.21 |
| NCNC | 95.42±2.41 | 48.68±18.60 | 86.49±0.99 | 20.31±6.51 | 81.86±1.64 | 35.67±12.30 |
| Bert | 35.79±2.50 | 3.42±0.47 | 36.12±0.37 | 6.56±0.70 | 37.66±1.57 | 10.04±0.85 |
| MiniLM | 83.39±0.00 | 34.29±4.10 | 66.35±0.29 | 21.54±0.11 | 68.15±0.09 | 16.91±0.18 |
| e5-large-v2 | 64.35±1.56 | 24.40±2.48 | 71.32±0.86 | 21.79±1.58 | 75.03±0.28 | 21.69±0.03 |
| Llama-3-8B | 89.15±0.72 | 31.19±3.49 | 79.87±1.19 | 22.87±4.47 | 83.18±1.19 | 22.85±1.12 |
| Llama-BUDDY | 91.23 ± 1.66 | 29.10 ± 5.72 | 88.10 ± 2.08 | 24.57 ± 3.66 | 75.15 ± 3.55 | 28.74 ± 5.29 |
| e5-large-BUDDY | 92.17 ± 1.32 | 34.19 ± 10.77 | 88.07 ± 0.57 | 22.90 ± 5.86 | 64.52 ± 3.00 | 20.79 ± 4.67 |
| BERT-BUDDY | 63.32 ± 5.74 | 14.64 ± 3.70 | 62.03 ± 3.16 | 13.15 ± 2.84 | 35.19 ± 1.54 | 10.20 ± 2.88 |
| MiniLM-BUDDY | 92.41 ± 1.62 | 41.05 ± 11.41 | 85.98 ± 1.19 | 25.45 ± 4.86 | 38.33 ± 2.89 | 11.31 ± 1.78 |
| Llama-NCN | 94.94 ± 0.85 | 49.62 ± 10.03 | 90.05 ± 1.66 | 30.73 ± 10.11 | 86.43 ± 0.24 | 32.48 ± 12.29 |
| e5-large-NCN | 95.57 ± 1.02 | 49.65 ± 14.61 | 90.97 ± 1.77 | 32.19 ± 4.80 | 84.36 ± 0.56 | 32.23 ± 14.04 |
| BERT-NCN | 73.04 ± 4.08 | 43.37 ± 8.81 | 67.95 ± 3.42 | 23.16 ± 6.75 | 61.93 ± 0.78 | 37.05 ± 8.86 |
| MiniLM-NCN | 98.58 ± 0.60 | 46.59 ± 11.01 | 88.85 ± 0.89 | 33.60 ± 7.51 | 61.95 ± 0.54 | 31.86 ± 13.16 |
| Llama-NCNC | 95.57 ± 1.02 | 27.45 ± 7.86 | 84.65 ± 1.95 | 20.51 ± 9.80 | 84.68 ± 1.72 | 27.16 ± 11.48 |
| e5-large-NCNC | 96.13 ± 1.13 | 39.23 ± 12.99 | 90.86 ± 1.95 | 27.02 ± 5.96 | 83.24 ± 1.20 | 25.14 ± 9.39 |
| BERT-NCNC | 77.47 ± 1.77 | 25.39 ± 12.42 | 72.80 ± 1.78 | 23.49 ± 3.07 | 58.83 ± 3.91 | 22.80 ± 2.55 |
| MiniLM-NCNC | 96.13 ± 1.20 | 38.96 ± 13.20 | 90.32 ± 1.52 | 22.56 ± 3.30 | 65.65 ± 1.80 | 29.10 ± 3.83 |
| Llama-NeoGNN | 77.63 ± 4.33 | 47.11 ± 8.51 | 64.00 ± 0.87 | 33.82 ± 3.81 | 56.20 ± 10.94 | 22.39 ± 13.38 |
| e5-large-NeoGNN | 78.91 ± 2.49 | 47.04 ± 8.63 | 64.74 ± 2.03 | 32.65 ± 5.49 | 63.56 ± 11.23 | 24.70 ± 11.08 |
| BERT-NeoGNN | 80.08 ± 4.22 | 45.50 ± 8.99 | 64.10 ± 1.50 | 33.01 ± 5.49 | 57.40 ± 9.79 | 23.92 ± 8.63 |
| MiniLM-NeoGNN | 81.35 ± 5.27 | 43.83 ± 6.82 | 65.13 ± 0.26 | 33.89 ± 6.18 | 62.53 ± 14.04 | 21.49 ± 18.99 |
| BERT-FT | 89.17±2.86 | 30.90±4.33 | 73.70±4.01 | 17.11±3.90 | 77.75±3.46 | 29.54±3.98 |
| e5-large-v2-FT | 92.09±1.70 | 38.63±9.39 | 76.26±2.55 | 19.75±5.81 | 80.48±2.52 | 31.73±6.62 |
| MiniLM | 92.49±2.13 | 35.55±5.82 | 75.87±3.72 | 20.79±6.32 | 80.20±2.62 | 29.86±5.82 |
| all-mpnet-base-v2 | 93.44 ± 1.64 | 22.55 ± 10.71 | 63.27 ± 31.76 | 9.38 ± 3.12 | 82.72 ± 1.28 | 8.42 ± 6.49 |
| MiniLM-GAT-CT | 54.23 ± 4.08 | 13.74 ± 5.21 | 29.44 ± 2.84 | 4.26 ± 1.75 | 13.76 ± 2.22 | 2.62 ± 0.55 |
| MiniLM-SAGE-CT | 63.04 ± 9.23 | 15.09 ± 1.35 | 45.31 ± 3.83 | 6.70 ± 3.61 | 49.11 ± 4.22 | 11.48 ± 3.41 |
| mpnet-GIN-CT | 89.01 ± 5.54 | 29.06 ± 7.96 | 46.82 ± 4.22 | 11.86 ± 3.68 | 55.11 ± 3.70 | 18.88 ± 5.89 |
| mpnet-GAT-CT | 74.90 ± 9.22 | 23.16 ± 11.10 | 35.82 ± 3.81 | 5.36 ± 1.69 | 19.50 ± 1.91 | 4.49 ± 0.91 |
| mpnet-SAGE-CT | 82.01 ± 4.19 | 25.34 ± 8.06 | 57.58 ± 3.78 | 11.91 ± 3.58 | 52.97 ± 5.05 | 14.51 ± 3.37 |
| MiniLM-LMGJOINT-C | 99.92±0.18 | 41.52±19.50 | 99.91 ± 0.09 | 44.99 ± 10.82 | 90.61 ± 2.25 | 35.47 ± 10.91 |
| MiniLM-LMGJOINT | 98.34±0.59 | 60.84±7.75 | 78.56 ± 6.32 | 22.72 ± 1.51 | 84.57 ± 1.93 | 40.27 ± 11.91 |
| mpnet-LMGJOINT-C | 93.28±14.16 | 28.92±7.14 | 99.27 ± 1.19 | 23.99 ± 11.63 | 73.09 ± 16.32 | 14.68 ± 6.17 |
| mpnet-LMGJOINT | 100.00±0.00 | 68.43±14.23 | 91.67 ± 4.96 | 31.66 ± 5.33 | 89.17 ± 5.45 | 45.70 ± 3.88 |
| e5-large-LMGJOINT-C | 99.92±0.18 | 41.46±25.49 | 99.11 ± 1.54 | 21.66 ± 9.66 | 83.97 ± 4.23 | 12.66 ± 2.66 |
| e5-large-LMGJOINT | 96.29±2.08 | 65.26±11.52 | 77.34 ± 2.19 | 23.80 ± 3.29 | 80.01 ± 2.53 | 42.02 ± 5.56 |