# OpenReview forum: "Link Prediction on Text Attributed Graphs: A New Benchmark and Efficient LM-nested GNN Design"
_ICLR.cc/2025/Conference — ICLR 2025 Conference Withdrawn Submission_

### Official Review · Reviewer_bDwZ · 2024-11-01

**Soundness:** 3
**Presentation:** 3
**Contribution:** 3
**Rating:** 6
**Confidence:** 3

**Summary:**

This paper presents a comprehensive benchmark for link prediction on text-attributed graphs (TAGs) and proposes a novel PLM-nested GCN design called LMGJOINT. The authors analyze the importance of PLMs in link prediction and investigate the representation power of text encoders. They introduce LMGJOINT, which includes residual connections of textual proximity, a combination of structural and textual embeddings, and a cache embedding training strategy. Experimental results show that LMGJOINT outperforms previous state-of-the-art methods and achieves competitive performance across multiple datasets.

**Strengths:**

(1) The paper provides an innovative and practical methodology for link prediction on TAGs, addressing the importance of PLMs and proposing a novel PLM-nested GCN design.

(2) The paper conducts an extensive benchmark to compare current competitive link prediction methods and PLM-based methods, providing a comprehensive evaluation of different approaches.

(3) The paper is well-structured and provides a clear overview of related work, datasets, methodologies, and experimental results.

**Weaknesses:**

(1) Edge Text Consideration. The dataset should incorporate edge text to enhance its robustness and relevance. For instance, in a citation network dataset, including the citation context as edge text would provide valuable semantic information, improving the analysis and interpretations of the relationships between nodes.

(2) Domain Limitation of Datasets. Expanding the evaluation to include diverse domains, such as those in the TEG-DB datasets [1], which feature rich node and edge text, would strengthen the findings.

(3 ) Writing Quality: There are several typos present, and the overall writing quality needs improvement for clarity and professionalism.

+ On page 1, Figure 1“X \in \mathbb{R}^d” should be “X \in \mathbb{R}^n*d”.
+ On page 1, Figure 1 caption “HC” should be “H_{c}”.

[1] "TEG-DB: A Comprehensive Dataset and Benchmark of Textual-Edge Graphs." NeurIPS 2024.

**Questions:**

Please see the Weaknesses above.

---

> ### Author Response · Authors · 2024-11-20
> **Immediate Response to Extensive Experiment Plan and Clarification for Edge-attributed TAG**
>
> Dear Reviewer and Area Chairs,
>
> Thank you for your commitment to reviewing our paper and suggestions to improve the work. Below are our detailed responses to your inquiries:
>
> (1) Edge Text Consideration. The dataset should incorporate edge text to enhance its robustness and relevance. For instance, in a citation network dataset, including the citation context as edge text would provide valuable semantic information, improving the analysis and interpretations of the relationships between nodes.
> Thank you for your suggestion to improve the generality of our method.
> We have started benchmarking two graphs from social and e-commerce domains from TEG_Benchmark[4] in Table 2. Which will be soon updated in our upload paper.
> From the perspective of research methodology, we considered merging the datasets with and without edge features in the early stage of the project. However, after conducting a literature review and building the baselines for this project, we decided to narrow our focus and concentrate on the contribution of text embeddings to link prediction without edge features.
> Our reasoning is as follows:
> The aggregation of edge features in MPNN has not yet been systematically studied and benchmarked. Our previous study of existing methods includes three categories:
> 1. Knowledge Graphs: KGEmbedding treats edge embedding as a vector in Euclidean space which can be modeled as different geometric transformations (e.g., translations, rotations) in the embedding space, such method includes RotatE[5] and TransE[6].
> 2. Biological Graphs: In material science, the most well-known work [2] uses the dot product of edge embeddings and node embeddings to predict graph class. It does not fulfill a standard MPNN paradigm.
> 3. Adjacency Matrix Construction:  GCN [3] and TEG-DB [4] utilize edge features to generate adjacency matrices, which guide the message flow in MPNN.
>
> Our work aims to explore the role of text embeddings and investigate effective design for fine-tuning strategy for TEG. We sincerely appreciate your valuable comments, which could potentially improve the generality of our method. But we believe TEG-DB leads to the departure from our objective of this work. We hope our motivation to focus on this subject and our clarification effectively addresses your concerns. However, we believe the reviewer's suggestion will serve as a good starting point for our next project.
>
> (2) Domain Limitation of Datasets. Expanding the evaluation to include diverse domains, such as those in the TEG-DB datasets [1], which feature rich node and edge text, would strengthen the findings.
> Thanks for your suggestion to improve the generality of our experiment design. We have started benchmarking two other graphs from the social and e-commerce domains in Table 2. If you have a specific requirement, please leave us a comment.
> We sincerely appreciate your valuable comments, which have provided us with new insights. We hope our response and experiment plan fulfill your expectations and effectively address your concerns.
>
> (3 ) Writing Quality: There are several typos present, and the overall writing quality needs improvement for clarity and professionalism.
> On page 1, Figure 1“X \in \mathbb{R}^d” should be “X \in \mathbb{R}^n*d”.
> On page 1, Figure 1 caption “HC” should be “H_{c}”.
> Thank you very much for your concrete suggestion about typos and clarification, we are in the process of improving the paper writing and clarification. We will incorporate all these issues in our updated pdf. Please wait patiently.
>
>
> [1] Wang, Yue et al. “Dynamic Graph CNN for Learning on Point Clouds.” ACM Transactions on Graphics (TOG) 38 (2018): 1 - 12.
> [2] Gilmer, Justin et al. “Neural Message Passing for Quantum Chemistry.” International Conference on Machine Learning (2017).
> [3] Defferrard, Michaël et al. “Convolutional Neural Networks on Graphs with Fast Localized Spectral Filtering.” Neural Information Processing Systems (2016).
> [4]  Li, Zhuofeng, et al. "TEG-DB: A Comprehensive Dataset and Benchmark of Textual-Edge Graphs." arXiv preprint arXiv:2406.10310 (2024)
> [5] Sun, Zhiqing et al. “RotatE: Knowledge Graph Embedding by Relational Rotation in Complex Space.” ArXiv abs/1902.10197 (2018): n. Pag.
> [6] Bordes, Antoine et al. “Translating Embeddings for Modeling Multi-relational Data.” Neural Information Processing Systems (2013).

---

> ### Author Response · Authors · 2024-11-27
> **Respond to limited domains**
>
> **Dear Reviewer bDwz and Area Chair,**
>
> Thank you for your valuable suggestion regarding evaluating our method on a broader domain. In response, we have conducted additional experiments using two datasets from TEG-DB, namely **History** and **Photo**. The results are presented in the table below:
>
> | **Dataset**        | **TF-IDF**          | **Word2Vec**        | **MiniLM-L6-v2**     | **e5-large-v2**      | **BERT**           | **Llama-3-8B**      | **NCN**            | **NCNC**           | **mpnet-FT**       | **e5-large-v2-FT**  | **MiniLM-L6-v2-FT** | **MiniLM-NCN**      | **e5-large-NCN**    | **MiniLM-\method**  | **mpnet-\method**   |
> |---------------------|---------------------|---------------------|----------------------|----------------------|--------------------|---------------------|--------------------|--------------------|--------------------|---------------------|---------------------|---------------------|---------------------|---------------------|---------------------|
> | **History**         | 60.94 ± 0.48 | 65.54 ± 0.21  | 94.64 ± 0.01   | 95.97 ± 0.00   | 90.26 ± 0.36 | 97.28 ± 0.08  | 97.77 ± 0.30 | 97.79 ± 0.25 | 93.91 ± 0.64 | 94.82 ± 1.18  | 95.58 ± 0.51  | 99.42 ± 0.11  | 99.50 ± 0.13  | 99.14 ± 0.02  | 99.54 ± 0.01  |
> | **Photo**           | 62.80 ± 0.62 | 64.71 ± 0.12  | 84.14 ± 0.03   | 85.04 ± 0.44   | 73.12 ± 0.76 | 88.62 ± 0.26  | 96.58 ± 0.20 | 96.79 ± 0.25 | 82.26 ± 0.92 | 86.26 ± 1.16  | 88.34 ± 0.75  | 99.59 ± 0.03  | 99.56 ± 0.04  | 99.58 ± 0.04  | 99.63 ± 0.02  |
>
> From these experiments, we observe that our proposed method consistently achieves superior performance, ranking as the best model on 8 out of 9 datasets for the link prediction task. This extends its effectiveness to both social and e-commerce graphs beyond the original evaluation domain.
>
> Thank you once again for your insightful feedback, which helped us strengthen our analysis.
>
> Best regards,
> Chen

---

### Official Review · Reviewer_eUjR · 2024-11-01

**Soundness:** 2
**Presentation:** 2
**Contribution:** 2
**Rating:** 3
**Confidence:** 5

**Summary:**

This paper develops a benchmark for comparing graph-based and PLM-based baselines in link prediction tasks based on eight proposed TAG datasets. Furthermore, it introduces a parameter- and memory-efficient fine-tuning method, LMGJOINT, which effectively integrates structural and textual information via GCN and PLM. Experimental results demonstrate the effectiveness and scalability of LMGJOINT, showing consistent performance improvements over compared baselines.

**Strengths:**

1. This paper introduces eight TAG datasets for link prediction and provides an empirical benchmark that enables fair comparisons between graph-based and PLM-based baselines.
2. This paper proposes a language model graph joint design framework, LMGJOINT, effectively combining structural and textual embeddings from GCN and PLM, respectively. Moreover, LMGJOINT adopts a cache embedding strategy, significantly reducing the computational cost.
3. This paper conducts extensive experiments on eight proposed datasets for link prediction, illustrating the effectiveness and scalability of LMGJOINT with minimal hyperparameter tuning required.

**Weaknesses:**

1. Disorganized writing and structure. The writing structure of this paper should be reorganized for improved clarity. For example, the content in "Related Datasets and Current Situation" in Section 3 could be integrated with "Related Benchmarks" in Section 2. Moreover, more detailed analyses of the eight proposed graph datasets for link prediction are needed. "Link prediction using GCN" in Section 2 should be reorganized based on different categories, incorporating some recent GCN-based methods from 2024, such as HL-GNN [1].
2. Ambiguous benchmarking setup. In Section 5, the authors should describe more implementation details, particularly regarding the proposed LMGJOINT, such as the chosen GCN and PLM. It is important to clarify these details to ensure a healthier benchmarking and determine what kind of benefits LMGJOINT brings. Furthermore, Sections 5.1 and 5.2 should clearly list all compared baselines.
3. Missing important baselines. To comprehensively evaluate LMGJOINT's performance in link prediction, many traditional random work-based and relation learning-based methods should be included, such as Metapath2Vec [2], ConvE [3], and ComplEx [4]. They can effectively capture flexible topological structures of graphs and understand relations between nodes. Additionally, some recent GNN and LLM + GNN baselines should be also considered, such as HL-GNN[1], BUDDY [5], TAPE [6], and GraphGPT [7].
4. Typographical Errors. There are some typos in this paper. For example, In line 069, $\textbf{H}C$ should be revised as $\textbf{H}^C$ consistent with $\textbf{H}^K$. In line 144, there is a reference error in "TAG_Benchmark (Yan et al., 2023)". In line 177, the caption of Figure 2 lacks a period after "..., see Appendix G". In line 216, "feature as shown in Equ 4" should be corrected to "feature as shown in Eq. 4."

**Reference**
[1] Heuristic Learning with Graph Neural Networks: A Unified Framework for Link Prediction, KDD, 2024.
[2] metapath2vec: Scalable representation learning for heterogeneous networks, KDD, 2017.
[3] Convolutional 2d knowledge graph embeddings, AAAI, 2018.
[4] Complex embeddings for simple link prediction, ICML, 2016.
[5] Graph neural networks for link prediction with subgraph sketching, ICLR, 2023.
[6] Harnessing Explanations: LLM-to-LLM Interpreter for Enhanced Text-Attributed Graph Representation Learning, ICLR, 2024.
[7] GraphGPT: Graph Instruction Tuning for Large Language Models, SIGIR, 2024.

**Questions:**

1. Could the author further clarify and compare the differences between the widely used OGB datasets for link prediction and the eight datasets proposed in this paper?
2. Could the authors discuss the efficiency of LMGJOINT in comparison to other baselines, including in terms of training/inference time and memory usage? Such an analysis would be valuable for demonstrating that LMGJOINT is both parameter- and memory-efficient.

---

> ### Author Response · Authors · 2024-11-25
> **Respond to missing details of LMGJoint**
>
> Dear Reviewer eUjR and Area Chairs,
>
> Thank you for your commitment to reviewing our paper and for your suggestions for improving the work. Below are our detailed responses to your inquiries:
>
> [1] Disorganised writing and structure. We will incorporate these suggestions in our updated pdf.
>
> 1.2 Moreover, more detailed analyses of the eight proposed graph datasets for link prediction are needed.  Thanks for your suggestion, we will provide a new section in Appendix to provide such details.
>
> 1.3 "Link prediction using GCN" in Section 2 should be re-organized based on different categories, incorporating some recent GCN-based methods from 2024, such as HL-GNN [1].
> Thank you for your thorough review. Given the significant overlap between link prediction methods using GCNs and other graph-based approaches, we have consolidated them into a single section and organized the discussion by category.
>
> [3] Missing important baselines.  Graph embedding: Metapath2Vec [2], ConvE [3], and ComplEx [4], HL-GNN[1], BUDDY [5], LLM-GCN:  TAPE [6], and GraphGPT [7].
> Thank you for your suggestion to enhance the baseline evaluation. We have started incorporating HL-GNN and BUDDY into the current benchmark. The result is reported here and will be updated in our paper. Due to the limited availability of resources and the absence of edge embedding implementations for Metapath2Vec [2], ConvE [3], and ComplEx [4], we regret that we are unable to provide results for these methods. However, we have successfully integrated four additional knowledge graph embedding methods into our benchmark. They are TransE, RotatE, ComplEx and DistMult.
>  However, TAPE is a prompt-based method primarily designed for node classification, which requires a transformation to adapt it for link prediction tasks. Similarly, GraphGPT also requires significant modifications to align with our experimental setup. Due to time constraints, we regret that we could not include results for these methods in this version of the paper. We acknowledge their relevance and will consider transformation and integrating them in future iterations.
>
>
> | Metric   | Hits@50       | Hits@100      | MRR          | AUC          |
> |----------|---------------|---------------|--------------|--------------|
> | Arxiv 2023 |
> | RotatE   | 24.76 ± 1.31  | 34.57 ± 0.84  | 5.65 ± 1.20  | 76.73 ± 0.44 |
> | TransE   | 21.27 ± 1.81  | 30.22 ± 1.96  | 5.11 ± 1.16  | 73.03 ± 1.04 |
> | ComplEx  | 22.33 ± 1.05  | 32.17 ± 2.38  | 5.16 ± 1.07  | 73.31 ± 0.61 |
> | DistMult | 21.32 ± 0.45  | 32.37 ± 1.62  | 5.02 ± 1.24  | 73.25 ± 0.39 |
> | Pubmed |
> | RotatE   | 26.10 ± 2.89  | 38.58 ± 2.59  | 4.41 ± 1.24  | 80.89 ± 0.40 |
> | TransE   | 38.58 ± 3.64  | 50.04 ± 2.63  | 7.74 ± 1.71  | 85.29 ± 0.97 |
> | ComplEx  | 38.14 ± 3.55  | 53.87 ± 3.09  | 6.50 ± 2.48  | 89.60 ± 0.51 |
> | DistMult | 39.09 ± 3.93  | 54.82 ± 2.95  | 6.90 ± 2.50  | 89.91 ± 0.53 |
> | Cora |
> | ComplEx  | 62.53 ± 1.71  | 73.59 ± 0.90  | 17.82 ± 2.58 | 75.71 ± 1.09 |
> | DistMult | 64.90 ± 1.59  | 73.12 ± 0.28  | 19.28 ± 2.11 | 76.53 ± 0.80 |
> | TransE   | 32.17 ± 6.52  | 57.00 ± 7.06  | 3.97 ± 0.90  | 64.19 ± 3.83 |
> | RotatE   | 27.51 ± 6.39  | 56.60 ± 6.29  | 2.98 ± 0.41  | 61.57 ± 3.88 |
>
> | Model   | Cora Hits@50   | Cora Hits@100  | Cora MRR      | Pubmed Hits@50 | Pubmed Hits@100 | Pubmed MRR   | Arxiv 2023 Hits@50 | Arxiv 2023 Hits@100 | Arxiv 2023 MRR   |
> |---------|----------------|----------------|---------------|----------------|-----------------|--------------|--------------------|---------------------|------------------|
> | BUDDY   | 87.82 ± 3.14   | 95.42 ± 2.26   | 30.78 ± 5.55  | 76.14 ± 3.46   | 89.25 ± 2.27    | 19.46 ± 2.42 | 52.25 ± 2.01       | 60.49 ± 0.94        | 18.75 ± 3.71     |
> | HL-GNN  | 90.59 ± 3.41   | 94.62 ± 1.87   | 50.35 ± 10.07 | 85.14 ± 1.83   | 91.87 ± 1.36    | 31.49 ± 7.84 | 76.51 ± 1.68       | 84.23 ± 0.82        | 24.21 ± 7.69     |
>
>
> Reference
> [1] Heuristic Learning with Graph Neural Networks: A Unified Framework for Link Prediction, KDD, 2024.
> [2] metapath2vec: Scalable representation learning for heterogeneous networks, KDD, 2017.
> [3] Convolutional 2d knowledge graph embeddings, AAAI, 2018.
> [4] Complex embeddings for simple link prediction, ICML, 2016.
> [5] Graph neural networks for link prediction with subgraph sketching, ICLR, 2023.
> [6] Harnessing Explanations: LLM-to-LLM Interpreter for Enhanced Text-Attributed Graph Representation Learning, ICLR, 2024.
> [7] GraphGPT: Graph Instruction Tuning for Large Language Models, SIGIR, 2024.

---

> > ### Comment · Reviewer_eUjR · 2024-11-26
> > **Response to authors' rebuttal**
> >
> > The authors have made efforts to respond to some of my comments. However, I still have some concerns, particularly regarding **Weakness 2** and **Questions 1-2**, which impact my evaluation of this paper's effectiveness and novelty.
> >
> > Therefore, I am maintaining my original rating for this paper.

---

> ### Author Response · Authors · 2024-11-26
> **Respond for Benchmark Setting**
>
> **Response to Reviewer eUjR and Area Chairs**
>
> Dear Reviewer eUjR and Area Chairs,
>
> Thank you for your thoughtful review of our paper and your valuable suggestions for improvement. Below, we provide detailed responses to your inquiries:
>
> ---
>
> ### **[2] Ambiguous Benchmarking Setup**
>
> In the proposed method, we employ the GCN architecture from [0]. Additionally, we benchmark three sentence embedding methods: **e5-large-v2** [1], **Sentence-Transformers MiniLM-L6-v2** [2], and **MPNet** [3]. To address your concerns, we have refined the relevant section in the paper as follows:
>
> ---
>
> #### **C2: Semantic Feature Proximity**
>
> In homophilic graph settings, connected nodes often exhibit high textual proximity. A straightforward approach in such cases is to disregard the graph structure and train a multilayer perceptron (MLP) solely on the textual encodings. Let $ \mathbf{T} $ represent the raw text, and $ \mathbf{X}^T $ the embedded textual features obtained from the pre-trained language model (PLM). We define the semantic proximity between node pairs $ (i, j) $ as:
> $\mathbf{X}^T = \text{PLM}(\mathbf{T})  $
> $ \mathbf{S}_{ij}^{T} = \mathbf{W}(\mathbf{X}^T_i \odot \mathbf{X}^T_j)  $
>
> Here, $ \mathbf{W} $ is a learned weight matrix, and $ \odot $ denotes the Hadamard product. The PLM maps raw text to a numerical vector representation. For benchmarking, we evaluate three PLMs: **e5-large-v2** \citep{Wang2022TextEB}, **Sentence-Transformers MiniLM-L6-v2** \citep{Reimers2019SentenceBERTSE}, and **MPNet** \citep{Song2020MPNetMA}.
>
> ---
>
> #### **C3: Aggregated Semantic Feature with Self-loop**
>
> To effectively capture the information of $k$-step neighbors, we aggregate features and propagate them through self-loops. This approach is particularly beneficial in homophilic settings, such as citation graphs \citep{lee2024netinfof}. The propagation mechanism is expressed as:
>
> $\mathbf{H}^{K} = f \left( \tilde{\mathbf{A}}_{\text{sym}}^K \mathbf{H}^0 \mathbf{W} \right) $
>
> Here, $ \mathbf{H}^0 = \mathbf{X} $, which is iteratively optimized using $ \mathbf{X}^T $ from the PLM, as described in the cache embedding strategy in Section \ref{subsec: efficient design}. The adjacency matrix is symmetrically normalized as:
>
> $ \tilde{\mathbf{A}}_{\text{sym}} = (\mathbf{D} + \mathbf{I})^{-\frac{1}{2}} (\mathbf{A} + \mathbf{I}) (\mathbf{D} + \mathbf{I})^{-\frac{1}{2}}  $
>
> where $ \mathbf{I} $ is the identity matrix and $ \mathbf{D} $ is the diagonal degree matrix.
>
> In the proposed method, we leverage GCN from \citet{Kipf2016SemiSupervisedCW}.
> We hope our clarifications and experiment updates meet your expectations and address your concerns comprehensively.
>
> [0] Kipf, Thomas and Max Welling. “Semi-Supervised Classification with Graph Convolutional Networks.” ArXiv abs/1609.02907 (2016): n. pag.
>
> [1]  Reimers, Nils and Iryna Gurevych. “Sentence-BERT: Sentence Embeddings using Siamese BERT-Networks.” Conference on Empirical Methods in Natural Language Processing (2019).
>
> [2] Reimers, Nils and Iryna Gurevych. “Sentence-BERT: Sentence Embeddings using Siamese BERT-Networks.” Conference on Empirical Methods in Natural Language Processing (2019).
>
> [3] Song, Kaitao et al. “MPNet: Masked and Permuted Pre-training for Language Understanding.” ArXiv abs/2004.09297 (2020): n. pag.

---

> ### Author Response · Authors · 2024-11-26
> **Respond for Benchmark Setting**
>
> Dear Reviewer eUjR and Area Chair:
>
> Thank you for your concrete question about the inner mechanism of the proposed method. We would like to clarify it with several ablation studies.
>
> [1] Ablation Study:
> We analyze the necessity of individual components in our proposed method. This study addresses the following questions:
>
> - **Q1**: Does the **Jumping Textual Connection (X_ij)** contribute to performance?
> - **Q2**: Are the **GCN layers** the main contributor to the observed improvements?
> - **Q3**: What is the relationship among the contributions of **C1 (CN module)**, **C2 (X_ij module)**, and **C3 (GCN module)**?
>
> ### **Performance on Pubmed**
> | **Metric**  | **Pubmed Hits@50** | **Pubmed Hits@100** | **Pubmed MRR** | **Pubmed AUC** |
> |--------------|---------------------|----------------------|----------------|----------------|
> | w. CN        | 72.16              | 88.27               | 14.12          | 97.93          |
> | w. X_ij      | 35.06              | 35.06               | 15.04          | 67.39          |
> | w. GCN       | 78.20              | 85.88               | 23.36          | 97.36          |
> | **Proposed** | **78.84**          | **88.90**           | **26.56**      | **98.26**      |
>
> ### **Performance on Cora**
> | **Metric**  | **Cora Hits@50** | **Cora Hits@100** | **Cora MRR** | **Cora AUC** |
> |--------------|-------------------|-------------------|--------------|--------------|
> | w. CN        | 95.26            | 98.81            | 39.17        | 96.16        |
> | w. X_ij      | 50.99            | 50.99            | 48.52        | 75.27        |
> | w. GCN       | 88.14            | 92.09            | 38.53        | 92.35        |
> | **Proposed** | **96.44**        | **99.60**        | **65.99**    | **97.38**    |
>
> ---
> 1. **Q1: Contribution of Jumping Textual Connection (X_ij)**
>    Results with **without X_ij (w. X_ij)** show significantly lower performance compared to the proposed method. This indicates that while X_ij alone is sufficient to improve performance, its integration into the full proposed method contributes to the significant performance gain, especially in terms of MRR.
>
> 2. **Q2: Importance of GCN Layers**
>    Aligning with the previous studies, removing GCN (w. GCN) highlights that GCN layers 's contribution to overall performance is data-dependent, in Cora, removing GCN layers leads to a sharp decline, particularly in terms of **MRR in Cora**. However, the **Proposed** method further improves these metrics by combining GCN with other components.
>
> 3. Q3: Relationship among C1 (CN), C2 (X_ij), and C3 (GCN)
>    - **CN Module**: Results with **w. CN** suggests that CN alone fails to provide a strong baseline, However, it effectively captures complex structural dependencies, leading to improved MRR.
>    - **X_ij Module**: The performance of **w. X_ij** reveals its strong enhancement for the performance, particularly in **Hits@100** and **AUC**. Nevertheless, it enhances the pairwise textual correlation of link prediction when combined with GCN and CN.
>    - **GCN Module**: The **w. GCN** variant highlights the variant and crucial role of GCN in different datasets, improving all metrics compared to CN or X_ij alone.
>
> **Conclusion**
> The Proposed method integrates the strengths of all components—CN, X_ij, and GCN—to achieve state-of-the-art performance across all metrics. The ablation study underscores the necessity of combining these modules to fully exploit both textual and structural information in link prediction tasks.
>
> We hope our response and experiment plan fulfill your expectations and effectively address your concerns.

---

### Official Review · Reviewer_ZPgD · 2024-11-03

**Soundness:** 3
**Presentation:** 3
**Contribution:** 3
**Rating:** 6
**Confidence:** 3

**Summary:**

The paper proposes a novel approach for link prediction on text-attributed graphs by integrating PLMs with GCNs. The authors introduce a new method called LMGJOINT, which optimizes the representation of both structural and textual information through residual textual proximity connections and a cache embedding training strategy. Additionally, the paper presents a comprehensive benchmark consisting of eight graphs to compare various existing and proposed methods, showing that LMGJOINT significantly outperforms the baselines in terms of MRR and other metrics.

**Strengths:**

1. The paper addresses a gap in the current research by proposing a joint framework that efficiently integrates PLMs with GCNs. This nested approach for link prediction on TAGs is novel and demonstrates strong performance improvements across multiple datasets.
2. The introduction of eight datasets, including well-established ones like Cora and PubMed, provides a robust experimental setup.
3. LMGJOINT is designed to be memory-efficient, making it scalable to large graphs.

**Weaknesses:**

1. One of the biggest concerns is the authors may not consider textual edge cases, which is quite essential in text-arrtibuted graph benchmarks.
2. The datasets used for evaluation, while comprehensive, are mostly academic in nature. The paper would benefit from demonstrating the applicability of LMGJOINT in more diverse, real-world scenarios (e.g., social networks or e-commerce systems) to show that the method generalizes well beyond citation networks.
3. The paper introduces a fine-tuning strategy for PLMs but does not delve deeply into its impact on performance compared to using frozen embeddings. It would be valuable to see more ablation studies or results that isolate the effect of fine-tuning versus other factors in the model's performance.

**Questions:**

More detailed ablation studies to isolate the impact of each design decision, especially the fine-tuning of PLMs, would clarify the relative contribution of each component of the model.

A minor issue: it would be more common to abbreviate "Equation" as "Eq." instead of "Equ".

---

> ### Author Response · Authors · 2024-11-20
> **Response to Extensive Dataset and Experiment Plan Suggestions**
>
> Dear Reviewer ZPgD and Area Chair:
>
> Thank you for your commitment to reviewing our paper and suggestions to improve the work. Below are our detailed responses to your inquiries:
>
> [W1] Extension to Edge-Attributed Graph
> Thank you for your suggestion to extend the TEG by including edge-attributed graphs. We would like to respectfully share your thoughts and our objective in this work first and then respond to this question directly.
> When including edge-attributed graphs into a dataset, it involves potentially extensive benchmarking of algorithms for handling edge features, TEG-DB [4] provides one of them. Such a benchmark, we believe, deviates from the objectives of our current paper. Our work aims to explore the role of text embeddings in link prediction and investigate effective hypotheses for future research. While the pairwise structure-based GCN is well understood, the importance of semantic embedding is neglected by the main link prediction community.
>
> We indeed considered merging these edge-attributed datasets. However, after conducting an early-stage literature review for this project, we decided to narrow our focus.  Our reasoning is as follows:
> The aggregation of edge features in MPNN has not yet been systematically studied and benchmarked. Our previous study of existing methods includes three categories:
> Knowledge Graphs: KGEmbedding treats edge embedding as a vector in Euclidean space which can be modelled as different geometric transformations (e.g., translations, rotations) in the embedding space, such method includes RotatE[5] and TransE[6].
> Biological Graphs: In material science, the first work [2] uses the dot product of edge embeddings and node embeddings to predict graph class.
> Adjacency Matrix Construction: GCN [3] and TEG-DB [4] utilize edge features to generate adjacency matrices (potentially a weighted graph), which guide the message flow in MPNN.
> Thus, we will evaluate our model on TEG-DB [4] dataset for improved generality however, we believe an edge-attributed text-attributed graph is beyond the focus of this work. We hope our motivation to focus on this subject or response effectively addresses your concerns. We sincerely appreciate the reviewer's suggestion, which we will seriously consider as the starting point for our next project.
>
> [W2] Experiment on extensive datasets
> Thank you for concrete suggestions to improve the generalizability of this proposed method. We have started our experiment on e-commerce and social graphs from TEG_Benchmark [4]. We will report the updated result in our Table 2 form in both the comment and updated paper.
>
>
> [D2] A minor issue: it would be more common to abbreviate "Equation" as "Eq." instead of "Equ".
> Thank you so much for your concrete suggestion to improve our paper writing. We will soon upload our updated paper.
>
> [1] Wang, Yue et al. “Dynamic Graph CNN for Learning on Point Clouds.” ACM Transactions on Graphics (TOG) 38 (2018): 1 - 12.
>
> [2] Gilmer, Justin et al. “Neural Message Passing for Quantum Chemistry.” International Conference on Machine Learning (2017).
>
> [3] Defferrard, Michaël et al. “Convolutional Neural Networks on Graphs with Fast Localized Spectral Filtering.” Neural Information Processing Systems (2016).
>
> [4]  Li, Zhuofeng, et al. "TEG-DB: A Comprehensive Dataset and Benchmark of Textual-Edge Graphs." arXiv preprint arXiv:2406.10310 (2024)
>
> [5] Chen, Zhikai et al. “Text-space Graph Foundation Models: Comprehensive Benchmarks and New Insights.” ArXiv abs/2406.10727 (2024): n. pag.

---

> > ### Author Response · Authors · 2024-11-26
> > **Ablation Study of Fine tune**
> >
> > Thank you for your commitment to reviewing our paper and suggestions to improve the work. Due to limited space in the manuscript, we present an additional ablation study below to analyze the necessity and impact of fine-tuning (FT) in our proposed method. Specifically, we compare the performance of three frameworks—PLM-MLP, PLM-NCNC, LMGJoint-C—with and without fine-tuning.
> >
> > | Module    | Cora Hits@50 | Cora MRR | Cora AUC | Pubmed Hits@50 | Pubmed MRR | Pubmed AUC |
> > |------------|-----------------|--------------|----------|----------------|------------|------------|
> > | wo FT BERT-MLP  |  35.79±2.50 | 56.90±3.26 | 3.42±0.47 | 36.12±0.37 | 48.73± 1.43 | 6.56±0.70 | 37.66±1.57 | 48.74± 1.15 | | |10.04±0.85 |
> > | w FT   BERT-MLP   | 89.17±2.86 | 96.99±1.36 | 30.90±4.33 | 73.70±4.01 | 84.45± 2.92 | 17.11±3.90|  77.75±3.46 | 87.56± 2.05 | 29.54±3.98 |
> > | wo FT - Mpnet-MLP | 89.15±0.72 | 95.64±0.41 | 31.19±3.49 | 79.87±1.19 | 89.01± 0.53 | 22.87±4.47 | 83.18±1.19 | 89.91± 0.19 | 22.85±1.12 |
> > | w FT - Mpnet-MLP | 93.44±1.64 | 97.78±0.66 | 22.55±10.71 | 63.27±31.76 | 90.69±2.49 | 9.38±3.12 | 82.72±1.28 | 91.44±0.75 | 8.42±6.49 |
> > | wo FT - NCNC | 96.13±1.20 | 98.81±0.49 | 38.96±13.20 | 90.32±1.52 | 96.11± 0.60 | 22.56±3.30 |
> > | w-FT LMGJOINT-C | 99.92±0.18 | 99.92±0.18 | 41.52±19.50 | 99.91±0.09 | 99.94±0.08 | 44.99±10.82 |
> >
> > Table 1: Fine-tuning ablation study: Impact of fine-tune on performance.
> >
> > **Findings and Observations**
> > Comparing the results for each framework (Lines 1–2, 3–4, and 5–6), fine-tuning consistently improves performance across metrics and datasets. For instance, the BERT-MLP framework's Cora Hits@50 increases from 35.79 to 89.17 and Pubmed Hits@50 rises from 36.12 to 73.70 after fine-tuning.
> >
> > **Conclusion**
> > This ablation study underscores the critical role of fine-tuning in unlocking the full potential of PLMs in link prediction tasks. T

---

> > > ### Comment · Reviewer_ZPgD · 2024-11-26
> > >
> > > Thanks for your thorough response. Based on the response and the scope of the manuscript, I'll keep the score as it is already positive.

---

> ### Author Response · Authors · 2024-11-25
> **Ablation Study**
>
> Dear Reviewer ZPgD and Area Chair:
>
> Thank you for your commitment to reviewing our paper and suggestions to improve the work. Due to the limited space, we past our ablation study as below:
>
> [Weakness 3, Question 1] Ablation Study:
>
> After presenting the overall performance on Cora and Pubmed, we analyze the necessity of individual components in our proposed method. This study addresses the following questions:
>
> - **Q1**: Does the **Jumping Textual Connection (X_ij)** contribute to performance?
> - **Q2**: Are the **GCN layers** the main contributor to the observed improvements?
> - **Q3**: What is the relationship among the contributions of **C1 (CN module)**, **C2 (X_ij module)**, and **C3 (GCN module)**?
>
> ### **Performance on Pubmed**
> | **Metric**  | **Pubmed Hits@50** | **Pubmed Hits@100** | **Pubmed MRR** | **Pubmed AUC** |
> |--------------|---------------------|----------------------|----------------|----------------|
> | w. CN        | 72.16              | 88.27               | 14.12          | 97.93          |
> | w. X_ij      | 35.06              | 35.06               | 15.04          | 67.39          |
> | w. GCN       | 78.20              | 85.88               | 23.36          | 97.36          |
> | **Proposed** | **78.84**          | **88.90**           | **26.56**      | **98.26**      |
>
> ### **Performance on Cora**
> | **Metric**  | **Cora Hits@50** | **Cora Hits@100** | **Cora MRR** | **Cora AUC** |
> |--------------|-------------------|-------------------|--------------|--------------|
> | w. CN        | 95.26            | 98.81            | 39.17        | 96.16        |
> | w. X_ij      | 50.99            | 50.99            | 48.52        | 75.27        |
> | w. GCN       | 88.14            | 92.09            | 38.53        | 92.35        |
> | **Proposed** | **96.44**        | **99.60**        | **65.99**    | **97.38**    |
>
> ---
> 1. **Q1: Contribution of Jumping Textual Connection (X_ij)**
>    Results with **without X_ij (w. X_ij)** show significantly lower performance compared to proposed method. This indicates that while X_ij alone is sufficient to improve  performance, its integration into the full proposed method contributes to the significant performance gain especially in terms of MRR.
>
> 2. **Q2: Importance of GCN Layers**
>    Align with previous study, removing GCN (w. GCN) highlight that GCN layers 's contribution to overall performance are data-dependent, in Cora, removing GCN layers leads to a sharp decline, particularly in terms of **MRR in Cora**. However, the **Proposed** method further improves these metrics by combining GCN with other components.
>
> 3. Q3: Relationship among C1 (CN), C2 (X_ij), and C3 (GCN)
>    - **CN Module**: Results with **w. CN** suggest that CN alone fails to provide a strong baseline, However, it effectivtly captures complex structural dependencies, leading to improved MRR.
>    - **X_ij Module**: The performance of **w. X_ij** reveals its strong enhancement for the performance, particularly in **Hits@100** and **AUC**. Nevertheless, it enhances the pairwise textual correlation of link prediction when combined with GCN and CN.
>    - **GCN Module**: The **w. GCN** variant highlights the variant and crucial role of GCN in different datasets, improving all metrics compared to CN or X_ij alone.
>
> **Conclusion**
> The Proposed method integrates the strengths of all components—CN, X_ij, and GCN—to achieve state-of-the-art performance across all metrics. The ablation study underscores the necessity of combining these modules to fully exploit both textual and structural information in link prediction tasks.

---

> ### Author Response · Authors · 2024-11-25
> **Respond to [Weakness 3, Question 1] Ablation Study**
>
> Dear Reviewer ZPgD and Area Chair:
>
> Thank you for your commitment to reviewing our paper and for your suggestions for improving the work. Due to the limited space, we paste our ablation study as below:
>
> [Weakness 3, Question 1] Ablation Study:
>
> After presenting the overall performance of Cora and Pubmed, we analyze the necessity of individual components in our proposed method. This study addresses the following questions:
>
> - **Q1**: Does the **Jumping Textual Connection (X_ij)** contribute to performance?
> - **Q2**: Are the **GCN layers** the main contributor to the observed improvements?
> - **Q3**: What is the relationship among the contributions of **C1 (CN module)**, **C2 (X_ij module)**, and **C3 (GCN module)**?
>
> ### **Performance on Pubmed**
> | **Metric**  | **Pubmed Hits@50** | **Pubmed Hits@100** | **Pubmed MRR** | **Pubmed AUC** |
> |--------------|---------------------|----------------------|----------------|----------------|
> | w. CN        | 72.16              | 88.27               | 14.12          | 97.93          |
> | w. X_ij      | 35.06              | 35.06               | 15.04          | 67.39          |
> | w. GCN       | 78.20              | 85.88               | 23.36          | 97.36          |
> | **Proposed** | **78.84**          | **88.90**           | **26.56**      | **98.26**      |
>
> ### **Performance on Cora**
> | **Metric**  | **Cora Hits@50** | **Cora Hits@100** | **Cora MRR** | **Cora AUC** |
> |--------------|-------------------|-------------------|--------------|--------------|
> | w. CN        | 95.26            | 98.81            | 39.17        | 96.16        |
> | w. X_ij      | 50.99            | 50.99            | 48.52        | 75.27        |
> | w. GCN       | 88.14            | 92.09            | 38.53        | 92.35        |
> | **Proposed** | **96.44**        | **99.60**        | **65.99**    | **97.38**    |
>
> ---
> 1. **Q1: Contribution of Jumping Textual Connection (X_ij)**
>    Results with **without X_ij (w. X_ij)** show significantly lower performance compared to the proposed method. This indicates that while X_ij alone is sufficient to improve performance, its integration into the full proposed method contributes to the significant performance gain, especially in terms of MRR.
>
> 2. **Q2: Importance of GCN Layers**
>    Aligning with the previous studies, removing GCN (w. GCN) highlights that GCN layers 's contribution to overall performance is data-dependent, in Cora, removing GCN layers leads to a sharp decline, particularly in terms of **MRR in Cora**. However, the **Proposed** method further improves these metrics by combining GCN with other components.
>
> 3. Q3: Relationship among C1 (CN), C2 (X_ij), and C3 (GCN)
>    - **CN Module**: Results with **w. CN** suggests that CN alone fails to provide a strong baseline, However, it effectively captures complex structural dependencies, leading to improved MRR.
>    - **X_ij Module**: The performance of **w. X_ij** reveals its strong enhancement for the performance, particularly in **Hits@100** and **AUC**. Nevertheless, it enhances the pairwise textual correlation of link prediction when combined with GCN and CN.
>    - **GCN Module**: The **w. GCN** variant highlights the variant and crucial role of GCN in different datasets, improving all metrics compared to CN or X_ij alone.
>
> **Conclusion**
> The Proposed method integrates the strengths of all components—CN, X_ij, and GCN—to achieve state-of-the-art performance across all metrics. The ablation study underscores the necessity of combining these modules to fully exploit both textual and structural information in link prediction tasks.

---

### Official Review · Reviewer_Jfry · 2024-11-03

**Soundness:** 2
**Presentation:** 2
**Contribution:** 2
**Rating:** 3
**Confidence:** 4

**Summary:**

This paper provides a benchmark for link prediction tasks on text-attributed graphs for models based on pre-trained language models (PLMs). This paper proposes a novel architecture for link prediction on Textual-Attribute Graphs (TAGs), introducing a "Residual Textual Proximity Connection" to enhance node representations by directly incorporating textual similarity information. The model aims to address over-smoothing in GNNs and claims to improve parameter and memory efficiency by leveraging smaller language models (LMs) as backbones.

**Strengths:**

1. The experiments cover multiple model configurations and report performance on various metrics.

2. The idea of ​​"residual connection of GCN" is proposed to solve the over-smoothing problem in GNN in text-attributed graph applications.

**Weaknesses:**

1. The readability of the article needs to be improved.

2. Some statments are confusing and not supported by exerimental results. Some unusual results are not discussed.

3. No dataset is provided in the referenced repository, hindering reproducibility. Some implementation details are missing.

**Questions:**

1. Please review the references carefully to avoid duplicate entries, such as [1-4]. Some closely related work on link prediction in textual-attribute graphs is missing; consider including relevant studies, such as [5].

2. Please thoroughly check for typos. For instance, in the caption of Figure 1, in line 67, "$Hij$" should be "$H_{ij}$"; in line 69, "$HC$" should be "$H_C$"; in line 869, "$2^10$" should appear as "$2^{10}$", etc.

3. How to intialize $H^k$ in Eq. (4)?

4. The concept of a residual connection in this paper is confusing. Why is adding textual proximity information to the GCN’s output "residual"? Additionally, the authors claim that this connection mitigates the over-smoothing issue, but no experimental evidence supports this. The GNN used does have more than 3 layers, which does not qualify as a "deep GNN."

5. In lines 256-257, the authors claim that the model is both parameter and memory-efficient. However, the experimental results do not substantiate this claim. In Table 2, MiniLM, MPNet, and e5-large are used as PLM backbones of the proposed model (parameters less than 500M). It is not challenging to fine-tune these models.

6. There are inconsistencies between Algorithm 1 and the model description in Section 4. For instance, what is SE-NCN? Where is $H^K$, $H_C$ and $H_T$ in Algorithm 1? What is $X^{\mathcal{G}}$ and where is $X^{\mathcal{G}}$ in Section 4?

7. How do the additional statistics assist with link prediction tasks? The authors list this as a contribution, but there is no discussion on its relevance. If they do not improve link prediction, what purpose do they serve?

8. Key implementation details are missing. For instance, what specific fine-tuning techniques were used for the language models? Was full-parameter fine-tuning or parameter-efficient fine-tuning employed? How is $\beta$ in Eq. (5) determined?

9. The caption for Table 2 indicates that boxed results represent the best performance, but this is not the case.

10. In Table 2, it is unusual that fine-tuning PLM + MLP (FT-PLM-MLP) performs significantly better than fine-tuning PLM + GNN (FT-PLM-GCN), which contrasts with the findings of most prior studies. This discrepancy deserves further discussion.

11. No datasets are provided in the repository referenced.

12. What is $\theta_{\mathcal{G},T}$ in Eq. (7)?

13. The assumptions in the proofs for Theorems A.1 and A.2 appear oversimplified

---
[1] Kipf, Thomas N., and Max Welling. "Semi-supervised classification with graph convolutional networks." arXiv preprint arXiv:1609.02907 (2016).

[2] Luan, Sitao, et al. "When do graph neural networks help with node classification? investigating the impact of homophily principle on node distinguishability." arXiv preprint arXiv:2304.14274 (2023).

[3] Mao, Haitao, et al. "Revisiting link prediction: A data perspective." arXiv preprint arXiv:2310.00793 (2023).

[4] Xu, Keyulu, et al. "How powerful are graph neural networks?." arXiv preprint arXiv:1810.00826 (2018).

[5] Li, Zhuofeng, et al. "TEG-DB: A Comprehensive Dataset and Benchmark of Textual-Edge Graphs." arXiv preprint arXiv:2406.10310 (2024).

---

> ### Author Response · Authors · 2024-11-27
> **Response to Question [5]**
>
> Dear Reviewer Jfry and Area Chair,
>
>
> Thank you for your commitment to reviewing our paper and suggestions to improve the work. Below are our detailed responses to your question 5.
>
> We appreciate the reviewer’s core question regarding the parameter and memory efficiency of our model. Compared to larger models like Llama3, the sentence embedding models we employ (e.g., MiniLM, MPNet, and e5-large) are indeed relatively compact, with fewer than 500M parameters. However, while fine-tuning these models, we frequently encountered out-of-memory (OOM) issues and instable improvement, which motivated our design choices. Below, we provide a detailed explanation:
>
> 1. **Complexity Analysis of Nested Architectures:**
>    The computational complexity of a nested architecture in our setup is approximately \(O(lhS^2N^2)\), where \(h\) is the hidden dimension of the text embedding, \(l\) is the number of layers, \(S\) is the token length, and \(N\) represents the number of nodes. When using large token lengths (e.g., an average of 500 tokens) and large numbers of nodes (e.g., 40,000 nodes), this complexity often results in OOM errors for A100 40gb RAM. Specifically, these errors are most prominent during key-value updates in the attention mechanism. Moreover, we observed that training quality with Llama3 in such scenarios without such design is notably unstable, which aligns with observed negative transfer in other work in link prediction [0].
>
> 2. **Our Proposed Cache Strategy:**
>    To address these challenges, we introduced a caching mechanism that optimizes memory usage by updating only the node features relevant to the current epoch. While this approach introduces a potential shift in the distribution of node weights, it consistently enhances the quality of text embeddings derived from sentence embedding methods. More importantly, it effectively mitigates OOM errors, enabling stable and efficient training.
>
>
> We greatly value your feedback and welcome further actionable suggestions for improving our work and its presentation.
>
> [0] Chen, Zhikai et al. “Text-space Graph Foundation Models: Comprehensive Benchmarks and New Insights.” ArXiv abs/2406.10727 (2024): n. pag.

---

> ### Author Response · Authors · 2024-11-27
> **Respond to Residual Connection Q4**
>
> Dear Reviewer Jfry and Area Chair:
>
> Q4: Thank you for your question regarding the presentation of the residual connection in our paper. We have updated the description in the revised PDF and would like to share the refined explanation below, incorporating the updated design:
>
> Design 1: Jumping Connection of Textual Similarity:  GNNs primarily performs node-level aggregation via summing the weighted neighboring features iteratively. Such a local smoothing mechanism helps generate more representative embedding when the homophily assumption holds [0]. However, it also smooths the rich semantic embedding from textual nuance during the smoothing process. We address such issues by combining pairwise semantic proximity without training at the last layer, i.e., the semantic similarity representations “jump” to the last layer [1]. A jump connection that bypasses the GCN, directly transmitting feature proximity (textual similarity) to the final embeddings, as illustrated in Figure 1 (b). Additionally, we provide a theoretical justification for this design from an information-theoretic perspective in Appendix A.1.
>
> [0] Luan, Sitao et al. “When Do Graph Neural Networks Help with Node Classification? Investigating the Impact of Homophily Principle on Node Distinguishability.” (2023).
>
> [1] Xu, Keyulu et al. “Representation Learning on Graphs with Jumping Knowledge Networks.” ArXiv abs/1806.03536 (2018): n. pag.
> We hope this clarifies our design rationale, and we are happy to address further questions or incorporate additional feedback to improve the presentation.

---

> ### Author Response · Authors · 2024-11-27
> **Respond to Q7 Graph Statistic**
>
> Dear Reviewer Jfry and Area Chair,
>
> Thank you for your insightful question, which addresses the fundamental motivation behind our dataset contribution. We believe further development of link prediction is hindered by the lack of efficient hypothesis and unthorough graph statistics representing the underlying data factor. We will continue working on this topic in another paper. From now on, we paste refined related paragraphs as below:
>
> Current Limitation: The limitations of applying GNNs for node classification on heterophily graphs are well understood. In comparison, prior works on GNN4LP are mostly based on handcrafted structure features. Despite the practical improvement, our understanding of the dominant data factor within GNN4LP remains incomplete. We identify three critical data factors: 1) Feature heterophily refers to the impact of dissimilar features for link prediction, i.e. discrepancies between feature proximity and structure [0]; 2) Structure hierarchy describes the hierarchical structure that widely exists in the citation network. When embedding such a graph in Euclidean space, GCN-based embedding incurs a large distortion compared to hyperbolic space [1]; 3) Pairwise local structure: This hypothesis originates from the intrinsic permutation invariance of GCN. This results in limited expressivity in distinguishing automorphic nodes [2]. To analyze and study their impact on link prediction from a data-centric perspective, we introduce 3 categories, including 12 graph statistics, as shown in Table 5. These statistics compactly and thoroughly quantify the above-mentioned three data factors. Our proposed dataset and statistics provide valuable resources to advance research in the TAG and GRL communities.
>
> We hope our clarifications and experiment updates meet your expectations and address your concerns comprehensively.
>
> In addition, we also sharpen our motivation in both the introduction and appendix.
> [0] Jiong Zhu, Gao Li, Yao-An Yang, Jinghua Zhu, Xuehao Cui, and Danai Koutra. On the impact of feature heterophily on link prediction with graph neural networks. ArXiv, abs/2409.17475, 2024a. URL https://api.semanticscholar.org/CorpusID:272910629.
> [1] Ines Chami, Rex Ying, Christopher Ré, and Jure Leskovec. Hyperbolic graph convolutional neural networks. Advances in neural information processing systems, 32:4869–4880, 2019. URL https://api.semanticscholar.org/CorpusID:202784587.
> [2] Benjamin Paul Chamberlain, Sergey Shirobokov, Emanuele Rossi, Fabrizio Frasca, Thomas
> Markovich, Nils Hammerla, Michael M. Bronstein, and Max Hansmire. Graph Neural Networks for Link Prediction with Subgraph Sketching, May 2023. URL http://arxiv.org/abs/2209.15486. arXiv:2209.15486 [cs].

---

> ### Author Response · Authors · 2024-11-27
> **Respond to Questions 1 2 3 9 12 13**
>
> Dear Reviewer Jfry and Area Chairs,
>
> Thank you for your commitment to reviewing our paper and insightful feedback. Below are our detailed responses to your inquiries:
>
> [W3]: 1. No dataset is provided in the referenced repository, hindering reproducibility.
> Thank you for highlighting the reproductivity issues. We’ve created a get-dataset.sh to download all datasets in bash. Our updated anonymous repository is {https://anonymous.4open.science/r/TAG4LP-2CA8/README.md}.
>
> Questions [1, 2, 9 ]: Thank you for your thoughtful suggestion, we have uploaded our updated paper with removed duplicate entries , rewritten reference, extended related dataset and benchmark.
>
> [3]: How to initialise in Eq. (4)?
> Node feature  $H_0$  is initialized by default node features  $X$ , and these pre-computed embeddings are stored in a cache. In the current mini-batch, we re-encode only the tokens associated with the target and source links $( X^T_i, X^T_j )$ using the PLM, and then concatenate them with the pre-computed node features as the input for the GCN:  $ X = [X_{V \setminus \{(i, j)\}}; X^T_i; X^T_j] $.  This approach significantly reduces the per-mini-batch computational cost from  $O(Nd)$  (where  N  is the number of nodes and  d  is the embedding dimension) to  $O(d)$ . Please refer to updated section 4 and Figure 1 for more detail.
>
> [8] Key implementation details are missing.
> Thank you for pointing this out. For all experiments, we employ full parameter fine-tuning, as we observed that parameter-efficient fine-tuning results in approximately a 2% performance drop without effectively addressing OOM errors in most scenarios. Therefore, we opted for full parameter fine-tuning to achieve better performance, particularly in terms of AUC. The value of $\beta$ is determined through hyperparameter tuning, with $\beta = 0.1$ being used consistently across all datasets. We have included this clarification in the Appendix.
>
> [12] The assumptions in the proofs for Theorems A.1 and A.2 appear oversimplified.
> Thank you for your thoughtful feedback. Theorem 1 is designed to justify the necessity of the jumping connection based on text similarity. We have clarified this design in our response to Residual Connection Q4. Given the time and space constraints of this paper, we will retain the simplified proof in its current form and aim to refine it in future work.
>
> Theorem 2 aims to investigate the relationship between MRR and Hits@1, which are often treated as independent evaluation metrics. Moreover, the underlying factors driving this correlation remain unclear. To address this, we have systematically analyzed the relationship between structure-based methods and MRR based on our empirical observation, as well as the connection between textual similarity and Hits@k. We observed that the predicted likelihood indeed follows a Gaussian distribution, where most mislabelled edges are located in the overlapping range. We believe this analysis provides a valuable perspective for objectively assessing the current evaluation framework.
>
> Concretely, we observed that MRR and Hits@1 demonstrate a strong practical correlation of 0.98 under our experimental settings. The proved theorem aligns closely with our results, and the Gaussian distribution assumption is well-supported by the observation. We would greatly appreciate any recommendations for relevant literature or suggestions for more realistic theoretical derivations, as these would help us further enhance our theoretical framework.

---

> ### Author Response · Authors · 2024-11-27
> **Respond to Q 10 & 12**
>
> Dear Reviewer Jfry and Area Chairs,
>
> Thank you for your insightful feedback and core question. Below are our detailed responses to your inquiries:
>
> [11] What is   in Eq. (7)?
> Here's an explanation of the terminology and concepts in the theorem 1:
> ### **Optimization Objective**
> - The optimization aims to find the parameter set $\theta $ that maximizes $P(Y | T, A) $, expressed as:
> $  \theta = \arg \max_{\theta_{G,T}} P(Y | T, A) $
> - $\theta_{G,T} $: The parameters to be optimized, which include:
>   - **$\theta_G $:** Parameters of the Graph Convolutional Network (GCN) responsible for encoding and processing topological features ($A $).
>   - **$\theta_T $:** Parameters of the Large Language Model (LLM) responsible for encoding textual features ($T $).
>
> [10] The PLM + MLP (FT-PLM-MLP) configuration significantly outperforms fine-tuning PLM + GNN (FT-PLM-GCN).
>
> Thank you for your insightful question. We shared a similar intuition at the beginning of this project. However, our initial experiments with fine-tuning PLM + GCN encountered challenges with negative transfer, consistent with observations reported in another benchmark study, TSGFM ([0], Section 4.2.2, Observation 2). Interestingly, this result link prediction contrasts with results in other benchmarks, such as node classification tasks ([0]).
>
> Through systematic analysis, we identified that performance improvements with PLM-based text embeddings are achievable under specific conditions:
> 1. **Incorporating pairwise textual similarity** into the final embeddings via concatenation.
> 2. **Fine-tuning pairwise textual similarity followed by a MLP** during training.
> These two findings ground our D1. Our extended results, presented in Table 8, demonstrate these findings. For instance, models like NCNC and BUDDY satisfy these conditions (D1), while models such as NeoGNN and SEAL do not. Consequently, we observed notable performance drops in NeoGNN, GCN, GIN, and GAT.
>
> To address these challenges, we proposed a method consisting of three components, with **C1: Jumping Connection of Textual Similarity** playing a pivotal role. To further illustrate its importance, we have included an ablation study that highlights the specific contribution of C1 to the overall performance.
>
> ### **Performance on Pubmed**
> | **Metric**  | **Pubmed Hits@50** | **Pubmed Hits@100** | **Pubmed MRR** | **Pubmed AUC** |
> |--------------|---------------------|----------------------|----------------|----------------|
> | w. CN        | 72.16              | 88.27               | 14.12          | 97.93          |
> | w. X_ij      | 35.06              | 35.06               | 15.04          | 67.39          |
> | w. GCN       | 78.20              | 85.88               | 23.36          | 97.36          |
> | **Proposed** | **78.84**          | **88.90**           | **26.56**      | **98.26**      |
>
> ### **Performance on Cora**
> | **Metric**  | **Cora Hits@50** | **Cora Hits@100** | **Cora MRR** | **Cora AUC** |
> |--------------|-------------------|-------------------|--------------|--------------|
> | w. CN        | 95.26            | 98.81            | 39.17        | 96.16        |
> | w. X_ij      | 50.99            | 50.99            | 48.52        | 75.27        |
> | w. GCN       | 88.14            | 92.09            | 38.53        | 92.35        |
> | **Proposed** | **96.44**        | **99.60**        | **65.99**    | **97.38**    |
>
> ---
> It is important to note that most prior improvements were achieved on node classification tasks, where the conditions and requirements differ from those of our current focus. Our approach specifically focus on link prediction, leveraging textual and structural features as well as fine-tune technology with stable training quality.
> We hope our clarifications and updated results address your concerns. If you have relevant literature to suggest, we would be happy to review it.
>
> [0] Chen, Zhikai et al. “Text-space Graph Foundation Models: Comprehensive Benchmarks and New Insights.” ArXiv abs/2406.10727 (2024): n. pag.
>
> [1] Chen, Zhikai et al. “Exploring the Potential of Large Language Models (LLMs)in Learning on Graphs.” ACM SIGKDD Explorations Newsletter 25 (2023): 42 - 61.

---

### Note · Authors · 2025-10-16

I have read and agree with the venue's withdrawal policy on behalf of myself and my co-authors.

---

### Meta-Review · Area_Chair_d55E · 2024-12-17

**Metareview:**

This manuscript suggests benchmarks for link prediction tasks on Text-Attributed Graphs (TAGs) for models based on pre-trained language models (PLMs). It proposes a parameter- and memory-efficient fine-tuning method named LMGJoint to integrate structural and textual information using GCN and PLM. A key design feature is introducing a residual connection of textual proximity and a cache embedding training strategy.

Reviewers have acknowledged the efforts in exploring multiple benchmarks and model configurations and introducing the idea of combining structural and textual embeddings from GCN and PLM. However, the issues raised by Reviewer Jfry and Reviewer eUjR have not been fully addressed, and the manuscript will benefit from re-evaluation after those issues are addressed. First, the authors should compare LMGJoint with other baselines regarding training/inference time and memory usage. Although one of the main claims of the proposed method is a parameter- and memory-efficient approach, such comparisons are missing. Second, authors should provide sufficient details about the proposed method, LMGJoint, and experiments (including baselines); the description should be enough for others to reproduce the results. Third, the authors' rebuttal to Reviewer Jfry is unclear, and it is hard to get the points.

Note: In response to Reviewer bDwZ, the authors revealed their names ("Chen"), which breaks the anonymity rule.

**Additional Comments On Reviewer Discussion:**

Please read the metareview. Some critical weaknesses and questions remain unaddressed, mainly raised by Reviewer Jfry and Reviewer eUjR. Reviewer eUjR explicitly mentioned, "However, I still have some concerns, particularly regarding Weakness 2 and Questions 1-2, which impact my evaluation of this paper's effectiveness and novelty." after reading the authors' rebuttal. Also, reviewers pointed out many confusing points, typos, and poor writing quality in the original submission, and the authors re-uploaded the updated version. Such major revisions may require a re-evaluation of the paper.

---

### Decision · Program_Chairs · 2025-01-22

Reject